# Cellular immunotherapy targeting CLL-1 for juvenile myelomonocytic leukemia

Juwita Werner [1,15,17], Alex G. Lee[1,17], Chujing Zhang[1], Sydney Abelson[1], Sherin Xirenayi[1], Jose Rivera[1], Khadija Yousuf[1], Hanna Shin[1], Bonell Patiño-Escobar [2], Stefanie Bachl[3,4], Kamal Mandal[2,16], Abhilash Barpanda[2], Emilio Ramos[2], Adila Izgutdina [2], Sibapriya Chaudhuri[5], William C. Temple [6,7], Shubhmita Bhatnagar[8], Jackson K. Dardis[8], Julia Meyer[1], Carolina Morales[1], Soheil Meshinchi[9], Mignon L. Loh [10], Benjamin Braun[1], Sarah K. Tasian [8,11], Arun P. Wiita [2,12,13,14,18] & Elliot Stieglitz [1,18] ✉

Juvenile myelomonocytic leukemia (JMML) is a myeloproliferative disorder that predominantly affects infants and young children. Hematopoietic stem cell transplantation (HSCT) is standard of care, but post-HSCT relapse is common, highlighting the need for innovative therapies. While adoptive immunotherapy with chimeric antigen receptor (CAR) T cells has improved outcomes for patients with advanced lymphoid malignancies, it has not been comprehensively evaluated in JMML. In the present study, we use bulk and single-cell RNA sequencing, mass spectrometry, and flow cytometry to identify overexpression of CLL-1 (encoded by *CLEC12A*) on the cell surface of cells from patients with JMML. We develop immunotherapy with CLL-1 CAR T cells (CLL1CART) for preclinical testing and report in vitro and in vivo anti-leukemia activity. Notably, CLL1CART reduce the number of leukemic stem cells and serial transplantability in vivo. These preclinical data support the development and clinical investigation of CLL-1-targeting immunotherapy in children with relapsed/refractory JMML.

Juvenile myelomonocytic leukemia (JMML) is an aggressive hematologic malignancy with myeloproliferative characteristics that affects young children and is associated with significant morbidity and mortality[1,2]. At presentation, infiltration of abnormally proliferating cells of the monocytic and granulocytic lineage frequently cause organ

dysfunction[3,4]. Pre-transplant treatment options for cytoreduction include hypomethylating agents and cytotoxic chemotherapy drugs[5]. Hematopoietic stem cell transplantation (HSCT) is curative in approximately half of patients with newly diagnosed JMML, but relapse or transformation to acute myeloid leukemia (AML) occurs in 40–50%

[1]Department of Pediatrics, Benioff Children's Hospitals, University of California, San Francisco, CA, USA. [2]Department of Laboratory Medicine, University of California, San Francisco, CA, USA. [3]Department of Medicine, University of California, San Francisco, CA, USA. [4]Gladstone-UCSF Institute of Genomic Immunology, San Francisco, CA, USA. [5]Division of Hematology/Oncology, Department of Medicine, University of California, San Francisco, CA, USA. [6]Division of Pediatric Allergy, Immunology, and Bone Marrow Transplant, University of California, San Francisco, CA, USA. [7]Division of Pediatric Oncology, University of California, San Francisco, CA, USA. [8]Division of Oncology and Center for Childhood Cancer Research, Children's Hospital of Philadelphia, Philadelphia, PA, USA. [9]Clinical Research Division, Department of Pediatrics, Fred Hutchinson Cancer Center, Seattle, WA, USA. [10]Seattle Children's Hospital, The Ben Towne Center for Childhood Cancer Research, University of Washington, Seattle, WA, USA. [11]Department of Pediatrics and Abramson Cancer Center, University of Pennsylvania School of Medicine, Philadelphia, PA, USA. [12]Department of Bioengineering and Therapeutic Sciences, University of California, San Francisco, CA, USA. [13]Chan Zuckerberg Biohub San Francisco, San Francisco, CA, USA. [14]Parker Institute for Cancer Immunotherapy, San Francisco, CA, USA. [15]Present address: Department of Pediatric Hematology and Oncology and Institute of Experimental Hematology, Hannover Medical School, Hannover, Germany. [16]Present address: Department of Animal Biotechnology, Gujarat Biotechnology University, Gandhinagar, India. [17]These authors contributed equally: Juwita Werner, Alex G. Lee. [18]These authors jointly supervised this work: Arun P. Wiita, Elliot Stieglitz. ✉e-mail: elliot.stieglitz@ucsf.edu

**Table 1 | Characteristics of the RNAseq, mass spectrometry, and flow cytometry cohorts**

| | Bulk RNAseq | Single cell RNAseq | Mass spectrometry | Flow cytometry |
|---|---|---|---|---|
| **Number** | 85 | 22 | 9 | 52 |
| **Median age at diagnosis (months)** | 20.7 | 24.8 | 9.1 | 20.3 (50 with data) |
| Range | 0.9–303.5 | 1–86.8 | 0.8–70.5 | 0.8–98.4 |
| **Gender** | | | | |
| Male | 51/83 (61.4%) | 16/22 (72.7%) | 4/9 (44.4%) | 30/51 (58.8%) |
| Female | 32/83 (38.6%) | 6/22 (27.3%) | 5/9 (55.6%) | 21/51 (41.2%) |
| **Elevated fetal hemoglobin for age** | 37/68 (54.4%) | 12/17 (70.6%) | 4/8 (50%) | 25/42 (59.5%) |
| **Median platelet count at diagnosis (10$^9$/L)** | 55 (75 with data) | 96.5 (20 with data) | 76.0 | 50 (44 with data) |
| Range | 5–297 | 15–223 | 26–161 | 8–223 |
| **Number of somatic mutations at diagnosis** | | | | |
| Mean | 1.4 | 1.8 | 1.4 | 1.4 (50 with data) |
| 1 | 56/85 (65.9%) | 10/22 (45.5%) | 6/9 (66.7%) | 34/50 (68%) |
| ≥2 | 27/85 (31.8%) | 12/22 (54.5%) | 3/9 (33.3%) | 16/50 (32%) |
| **Methylation category** | | | | |
| Low | 32/59 (54.2%) | 6/17 (35.3%) | 4/8 (50%) | 13/29 (44.8%) |
| Intermediate | 8/59 (13.6%) | 2/17 (11.8%) | 0/8 (0%) | 1/29 (3.4%) |
| High | 19/59 (32.2%) | 9/17 (52.9%) | 4/8 (50%) | 15/29 (51.7%) |

Clinical, laboratory, and sequencing information of JMML samples used in this study.

of patients[6]. Children who relapse post-HSCT have a 2-year overall survival of ~10% without second transplantation[7]. Innovative strategies to treat patients with this high-risk leukemia are therefore needed[8].

Adoptive cellular immunotherapy with chimeric antigen receptor (CAR) T cells has been remarkably effective in patients with relapsed/refractory B-lymphoid malignancies, including acute lymphoblastic leukemia (ALL), diffuse large B-cell lymphoma, and multiple myeloma, but has not been explored extensively in JMML[9,10]. A prior report demonstrated in vitro activity using CAR T cells targeting the granulocyte-macrophage colony-stimulation factor (GM-CSF) receptor against CD34$^+$ JMML cells[11]. Preclinical studies of CD123-directed CAR T cells (CD123CART) and CD33CART immunotherapies, both well-known AML targets, also demonstrated activity in an AML patient-derived xenograft (PDX) model that transformed from JMML[12,13]. However, comprehensive assessment of immunotherapy targets in JMML has been lacking to date.

Leukemia stem cells (LSCs) have been shown to drive relapse and progression in JMML and include hematopoietic stem and progenitor cells (HSPCs), such as Lin$^-$CD34$^+$CD38$^-$ hematopoietic stem (HSCs) and Lin$^-$CD34$^+$CD38$^+$ granulocyte-monocyte progenitor cells (GMPs)[14]. We therefore sought to develop cellular immunotherapy against JMML by employing a multi-modal omics strategy, focusing specifically on targeting chemoresistant LSCs. Bulk RNAseq and flow cytometry of primary JMML mononuclear cells (MNCs) revealed that *CLEC12A*/CLL-1 is overexpressed at both transcript and protein levels. Through single-cell (sc)RNAseq and mass spectrometry analysis, we further identified that CLL-1 can be upregulated on LSCs isolated from patients with JMML compared to cells from healthy controls.

C-type lectin-like receptors play an essential role in regulating innate and adaptive immunity[15]. CLL-1 recognizes various ligands on dead cells[16] and specifically recruits inhibitory Src homology region 2 domain-containing phosphatase (SHP)-1 and SHP-2, alleviating inflammation[17]. C-type lectin-like molecule-1 (CLL-1), encoded by *CLEC12A*, has emerged as an effective target in the treatment of AML, given its high expression on bulk AML cells and LSCs[18,19]. In normal human hematopoietic tissues, CLL-1 is highly expressed on myeloid progenitor cells[20] but is largely absent in normal Lin$^-$CD34$^+$CD38$^-$ HSCs[20], offering a unique therapeutic window for targeting myeloid leukemia[18,21].

CLL-1 CAR T cells have shown promise against AML in vitro and in vivo[18,22]. Clinical trials have indicated the tolerability and efficacy of CLL-1 CAR T cell therapy in children[23–26] and adults[27] with relapsed or refractory AML.

In this work, we observe potent in vitro killing of primary JMML cells by CLL-1 CAR T cells (CLL1CART). Importantly, we show that CLL1CART effectively inhibit in vivo leukemia proliferation in murine peripheral blood (PB) and end-study spleen and bone marrow (BM) tissues in JMML PDX models and specifically reduce LSCs and the ability to serially transplant JMML cells. We propose that CLL1CART immunotherapy could be used as salvage therapy in patients with relapsed/refractory JMML, potentially as a conditioning strategy for subsequent allogeneic HSCT.

## Results

### CLL-1 is upregulated on JMML MNCs

We first sought to define the JMML surface proteome ("surfaceome") to identify upregulated tumor cell surface antigens as potential targets for CAR T cell immunotherapy. We generated bulk RNAseq data from 85 primary samples (BM or PB MNCs) and compared it to normal pediatric MNCs and cord blood (CB) Lin$^-$/CD34$^+$-sorted cells[14] (Table 1 and Supplementary Data 1, 2, and 13). We filtered for genes encoding surface proteins using the Human Protein Atlas, a mass spectrometry-derived cell surface protein atlas[28] and the Cancer Surfaceome Atlas[29], and identified 1500 surface proteins expressed at the transcript level in this dataset. Using publicly available datasets for pediatric AML and ALL[30], we found that JMML patients have a distinct "surfaceome"-encoding gene set (Supplementary Fig. 1A, B).

We next pursued a bioinformatic strategy to identify potential therapeutic targets. We first classified 1310 surface protein-encoding genes as enriched in JMML compared to normal pediatric controls (Fig. 1A). To reduce the likelihood of "on-target, off-tumor" toxicity, from this list we excluded protein-encoding genes with a median transcript per million (TPM) > 20 on any non-hematopoietic tissue per the genotype-tissue expression database (GTEx). This filtering led to 224 surface proteins of interest. Finally, fifteen surface protein-encoding genes remained using a stringent log2 fold change (log2FC) cutoff for transcript expression of >6.5 between JMML and healthy donors (Fig. 1B). This final list included CLL-1 (encoded by *CLEC12A*) (Fig. 1C and Supplementary Fig. 1C), which was notable because this gene is known to be restricted to myeloid lineages during normal hematopoiesis and to be absent on healthy HSCs[18].

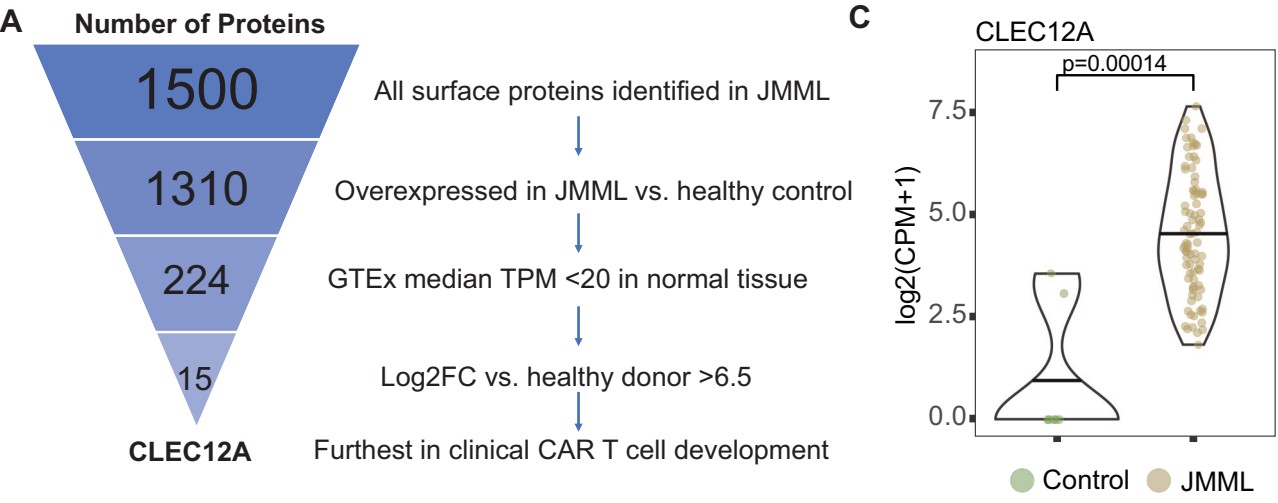

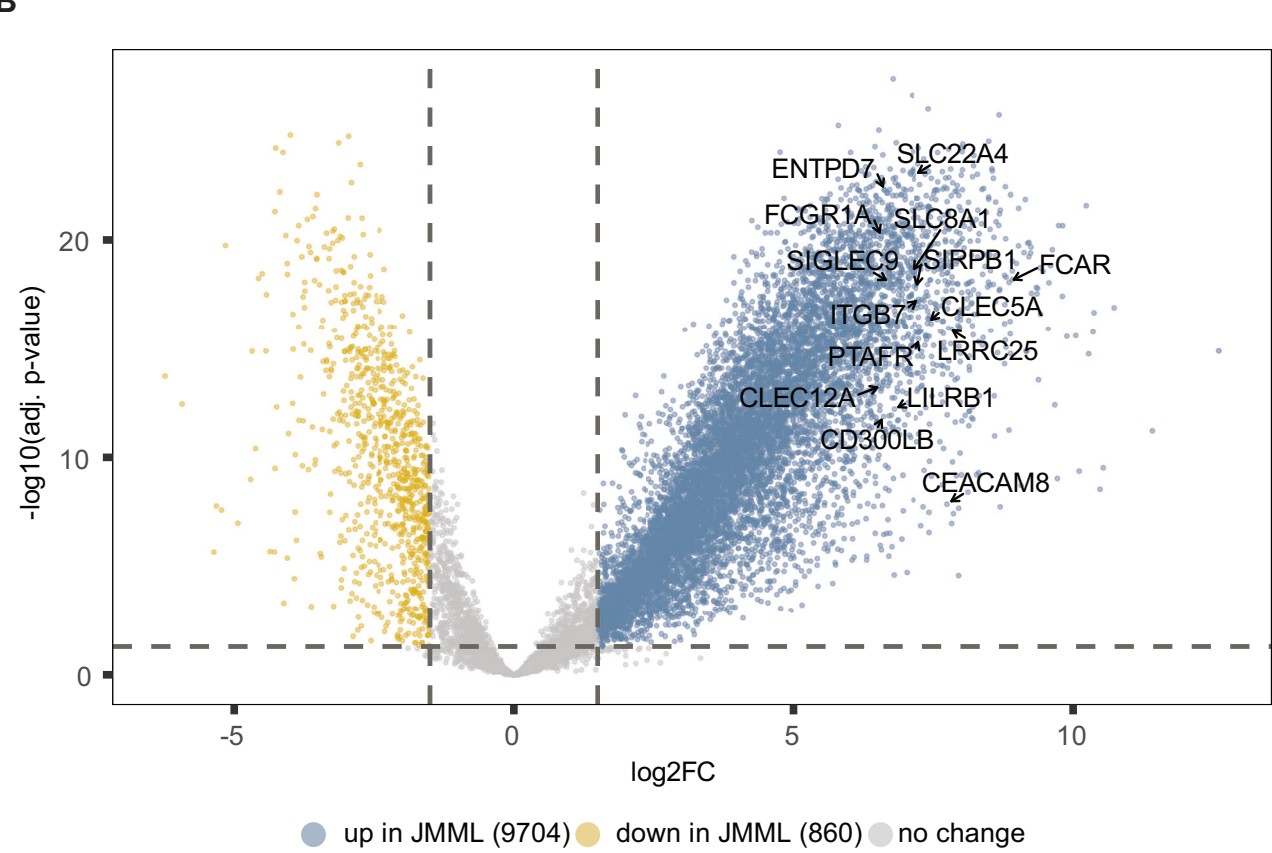

**Fig. 1 | Bulk RNAseq reveals *CLEC12A*/CLL-1 as a highly abundant cell surface gene/protein upregulated on JMML. A** Schematic showing triage of cell surface protein-encoding genes derived from bulk RNAseq (*n* = 85 for JMML, *n* = 7 for healthy controls) analysis to identify adoptive immunotherapy targets for JMML. Overexpression defined by adjusted p-value < 0.05. GTEx cutoff applies to all tissues except the hematopoietic tissues whole blood, spleen and EBV-transformed lymphocytes. **B** Volcano plot of identified gene transcripts from bulk RNAseq displaying up- or downregulated genes in JMML (log2FC > 1.5; adjusted *p*-value < 0.05). The 15 surface protein-encoding genes with log2FC > 6.5 and low GTEx expression are labeled. All statistics were derived from the linear model using the limma-voom package. **C** Log2(CPM + 1) from bulk RNAseq for *CLEC12A* comparing JMML and healthy controls. adj. *p*-value adjusted *p*-value, log2FC log2 fold change.

We observed uniform upregulation of *CLEC12A* across methylation groups (Supplementary Fig. 1D).

## CLL-1 can be overexpressed on JMML LSCs

We next generated scRNAseq data from 22 JMML MNC samples and compared them to corresponding data from four pediatric healthy control samples (age ≤ 10 years) (Table 1 and Supplementary Data 3, 4,

and 13). We used a similar approach to identify cell surface protein-encoding genes in our scRNAseq data as described for the bulk RNA-seq data, comparing the HSPC clusters (including LSC populations of interest such as HSCs and GMPs) of JMML versus healthy controls (Fig. 2A, B and Supplementary Fig. 2A, B). *CLEC12A* was overexpressed in the HSPC cluster, as well as HSCs and GMPs (Supplementary Fig. 2C, D). Pathway analysis of differentially expressed surface

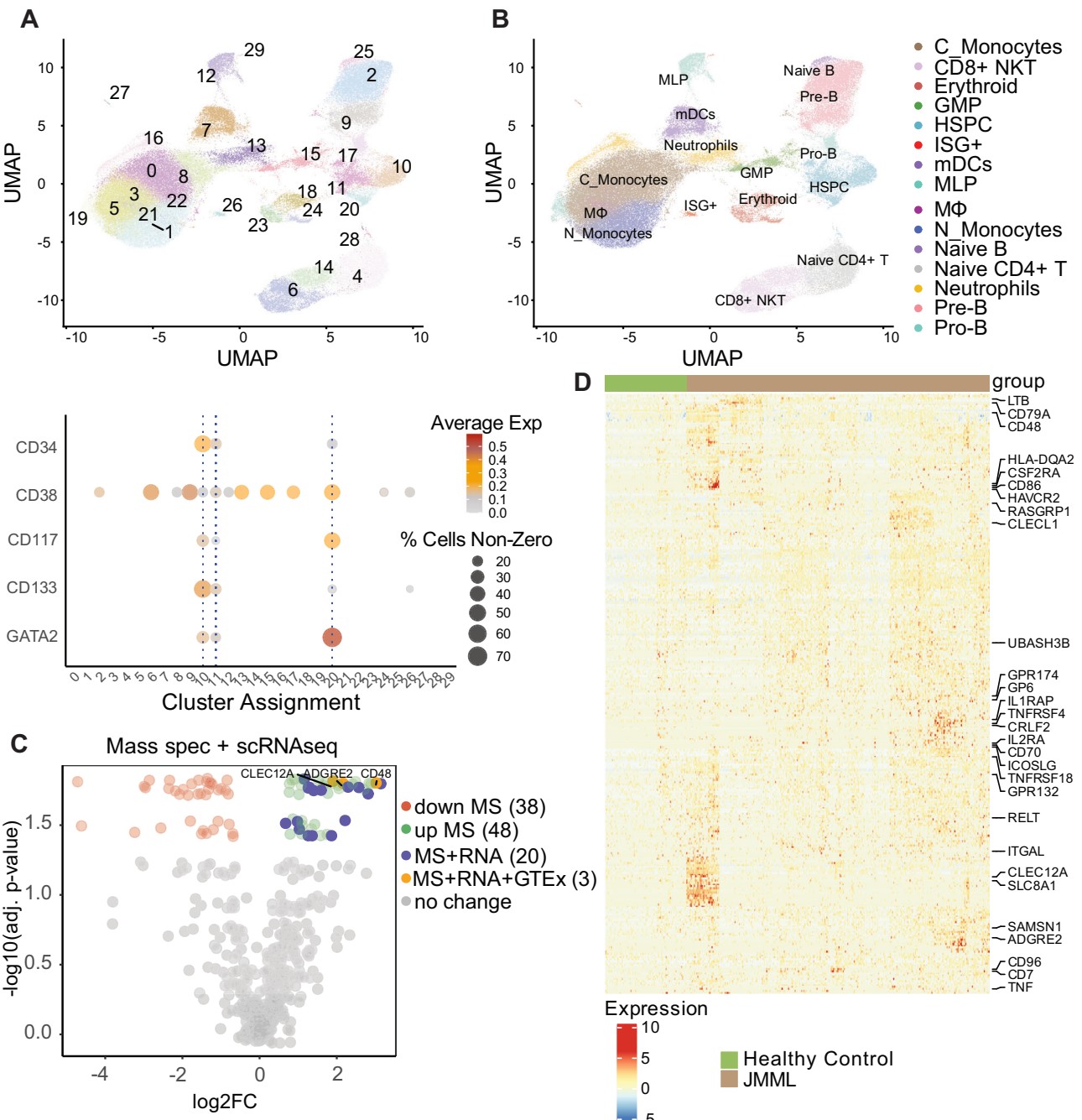

**Fig. 2 | Identifying CAR T cell targets on JMML LSCs by scRNAseq and mass spectrometry. A** Top: UMAP depicting assigned clusters. Bottom: the average expression of HSC markers in different assigned clusters. *Y*-axis shows genes while circle size and colors represent percent of cells expressing the gene or expression level, respectively. **B** UMAP with cell type identification overlay. **C** Volcano plot intersecting mass spectrometry of JMML ($n = 9$) vs. healthy control ($n = 3$) CD34+ cells with upregulated surface proteins highlighted (positive log2FC, FDR < 0.05; Wilcoxon signed-rank test with Benjamini–Hochberg correction). "MS + RNA" denotes significant proteins and genes upregulated by both mass spectrometry and scRNAseq, respectively. "+GTEx" denotes genes/proteins with median TPM <

20 in the GTEx database (except hematopoietic tissues). **D** Heatmap of differentially upregulated genes in JMML vs. normal control of the HSC cells using the FindMarkers function with the Wilcoxon rank sum test. Only cell surface and upregulated genes (FDR < 0.05) are shown. Genes associated with cell surface markers are labeled to the right. Hierarchical clustering of all HSC cells and their respective group assignment are labeled on top. Genes with TPM < 5 on at least 75% of all normal GTEx samples are further annotated to the right. Average Exp average expression, C_Monocytes classical monocytes, mDCs myeloid dendritic cells, N_Monocytes non-classical monocytes, ISG+ ISG expressing immune cells, MS/ Mass Spec mass spectrometry, RNA scRNAseq.

proteins indicated enrichment of proteins involved in cell activation, cytokine signaling, and adhesion of JMML HSPCs (Supplementary Fig. 2F and Supplementary Data 5).

To complement these transcriptomic results, we performed mass spectrometry-based proteomics on JMML CD34⁺ ($n = 9$) and adult healthy donor CD34⁺ HSCs ($n = 3$) (Table 1 and Supplementary Data 6,

7, and 13). Filtering for an adjusted *p*-value < 0.05 and a positive log2FC (JMML higher than healthy) in the scRNAseq and mass spectrometry data, there were 305 genes/proteins in the scRNAseq and 48 in the mass spectrometry dataset, of which 20 were overlapping. After selecting for proteins with a median TPM < 20 in normal tissue (except whole blood, spleen, EBV-transformed lymphocytes) in the GTEx

database, 33 genes and 6 proteins remained from the scRNAseq and mass spectrometry datasets, respectively.

Multiple C-type lectins and their ligands (such as *CLEC7A*, *CLEC10A*, and *CLECL1*), *CD7*, and *CD96* were upregulated in the scRNAseq data only. CD7 was previously described to be aberrantly expressed on CD34[+] cells in JMML[31], and there are both pre-clinical data[32] and early-phase clinical trials (NCT05377827) of CD7CARTs for patients with T-ALL or AML. CD96 was previously reported to be differentially overexpressed in JMML LSCs[14]. CD38 was identified as upregulated in the mass spectrometry data. CD38 is an established target in multiple myeloma[33] and is under pre-clinical and clinical investigation for some leukemias[34] and lymphomas[35]. CD38 can be targeted by CD38 monoclonal antibodies such as approved daratumumab and isatuximab, bispecific antibodies, and CAR T cells[36–38]. Interestingly, all-trans retinoic acid has been shown to drive enhanced CD38 expression of AML cells[39,40] and multiple myeloma[41,42], a phenomenon that we recently reported in JMML[43].

Only CD48, ADGRE2, and *CLEC12A*/CLL-1 were common to both datasets (Fig. 2C, D and Supplementary Fig. 2G, H). We focused our subsequent experimental studies upon targeting CLL-1 given our results showing its overexpression on JMML MNCs and HSPCs.

## Flow cytometry validates CLL-1 overexpression in JMML

Due to potential discrepancies between transcriptome-proteome correlation[44], we then sought to validate our transcriptomic results by flow cytometry using BM and PB MNCs from JMML patients. Correlation analysis between bulk RNAseq and mass spectrometry demonstrated a significant positive correlation between the two methods (Supplementary Fig. 3A, B).

CLL-1 expression was predominantly detected on both monocytic and granulocytic cells based on size and granularity characteristics (Supplementary Fig. 4A and Supplementary Data 8 and 13). We also compared CLL-1 to CD33 as a positive control given its broad cell surface expression in AML and clinical use of CD33-targeting antibody-based immunotherapies[45,46]. We confirmed higher CLL-1 expression in the MNC population based on a metric of median fluorescence intensity (MFI) > 1.5-fold increased versus an FMO (fluorescence minus one) CLL-1 control with a CLL-1 isotype antibody (Fig. 3A and Supplementary Fig. 4B, C) in 39 of 46 samples. We found CLL-1 and CD33 antigens to be expressed at similarly high levels on primary JMML cells (Fig. 3B). We also observed higher CLL-1[+] percentages in JMML PB (mean 51.96%) and BM cells (mean 50.7%) compared to normal PB and BM cells, respectively (Fig. 3C), with high inter-patient heterogeneity. Quantification of CLL-1 molecules on the surface of JMML MNCs revealed CLL-1 densities (Fig. 3D) consistent with the reported antigen threshold for lysis and cytokine production[47].

We also investigated CLL-1 overexpression in JMML versus healthy control HSPCs by flow cytometry. We observed that the percentage of CD34[+]CD38[−] and CD34[+]CD38[+] cells was significantly higher in PB MNCs of JMML patients compared to healthy controls (Supplementary Fig. 4D), in line with previous reports[31], and higher CLL-1 expression in PB JMML CD34[+] cells compared to healthy control CD34[+] cells (Fig. 3E). We observed a trend towards higher CLL-1 expression on CD34[+] and CD34[+]CD38[+] JMML BM cells versus controls (Fig. 3E and Supplementary Fig. 4E, F). On CD34[+]CD38[−] cells, CLL-1 expression was more heterogeneous compared to healthy controls (Supplementary Fig. 4E). Overall, expression of CLL-1 was not significantly different compared to control HSPCs, and CLL-1 was expressed on a minority of JMML cells in the CD34[+] compartment. Nevertheless, we pursued CLL-1 as a CAR T cell target for JMML as it provides the possibility of improved safety over other targeted therapies (like CD33 and CD123) because of its reduced expression on normal HSCs[48] (Supplementary Fig. 5).

## CLL1CART have anti-JMML activity in vitro

We next evaluated the preclinical activity of four newly-created second generation lentiviral CLL1CART (CD8α hinge (H) and transmembrane domain (TM) with 4-1 BB co-stimulatory domain (co-stim), IgG4 H + CD28 TM + CD28 co-stim, CD28 H + TM + co-stim, CD28 H + TM + 4-1BB co-stim) with an enhanced green fluorescent protein (GFP) reporter gene to monitor transduction efficiency (Fig. 4A). Human CLL-1[+] cell lines and primary JMML cells were used in vitro for selection of a best-in-class product for further experimental study.

We assessed the four CLL-1 CAR constructs in healthy donor T cells. A luciferase-based cytotoxicity assay using U937 histiocytic lymphoma cells demonstrated that CLL1CART containing a CD8α H + TM and 4-1BB co-stim evoked more effective anti-leukemia activity than the other three constructs compared to empty CAR (same CAR backbone as anti-CLL-1 CAR but without an antigen recognition domain) (Fig. 4B). This backbone also demonstrated efficacy in an Incucyte-based cytotoxicity assay against U937 and HL-60 acute promyelocytic leukemia cells, expressing CLL-1 at different intensities, and at different effector to target cell (E:T) ratios compared to empty CAR or untransduced (UTD) T cell negative controls (Supplementary Fig. 6A, B). We also confirmed target-specific killing, as there was no cytotoxicity against CLL-1-negative K562 chronic myeloid leukemia cells (Supplementary Fig. 6C) or A-375 melanoma cells (Supplementary Fig. 6D).

Next, we assessed cytotoxicity of our four CLL1CART against primary JMML cells in vitro using three biological replicates. All CLL1CART effectively killed myelomonocytic CD11b[+] or CD14[+] primary JMML cells in vitro (Fig. 4C). Both CLL1CART containing a 4-1BB co-stim were more persistent after 48-h co-culture with primary JMML MNCs compared to the CD28 co-stim, consistent with prior reports[49], and the one containing a CD8α H + TM domain demonstrated increased proliferation (Fig. 4D). We therefore continued subsequent experiments with a backbone containing a CD8α H + TM and 4-1BB co-stim (Supplementary Fig. 6E), which is also used in the FDA-approved anti-CD19CART tisagenlecleucel.

We next compared CLL1CART and CD33CART activity against primary JMML MNCs in vitro. We again observed effective killing of myeloid JMML cells, and there was no difference between the two CART in the eradication of CD11b[+] JMML cells as assessed by flow cytometry after co-incubation with CART (Fig. 4E). We also observed activity of CLL1CART at various E:T (Fig. 4F).

In line with the observed partial overexpression of CLL-1 on JMML LSCs, we demonstrated cytotoxicity of CLL1CART against CD34[+], CD34[+]CD38[−], and CD34[+]CD38[+] cells compared to UTD T cells in vitro (Fig. 4G). Taken together, our data demonstrate CLL-1 as an ideal target on JMML MNCs and potentially LSCs.

## T cell receptor alpha chain constant (TRAC) knock out reduce alloreactivity against primary JMML cells

CLL1CART demonstrated improved activity over control empty CAR and UTD T cells in vitro. However, we observed a known phenomenon of tonic CAR signaling and alloreactivity in vitro with empty CAR and UTD T cells having more anti-leukemia activity compared to no T cells (Fig. 4C, E, and F). We hypothesized that this non-specific cytotoxicity was due to T cell receptor (TCR)-mediated alloreactivity. We therefore used CRISPR/Cas9 gene editing to knock out the *TRAC* gene[50] in CLL1CART (CD8α H + TM, 4-1BB co-stim) and tested their in vivo anti-leukemia activity against a bioluminescent U937 cell line xenograft model to assess whether the suspected TCR-mediated alloreactivity was specific to primary cells. We did not observe TCR-mediated alloreactivity against U937 (Fig. 5A), and animal survival was similar between unmanipulated and *TRAC*-knockout (KO) cells within treatment groups (Fig. 5B, C). In a separate study with unmanipulated cells, we did not observe CLL-1 downregulation on remaining U937 cells in

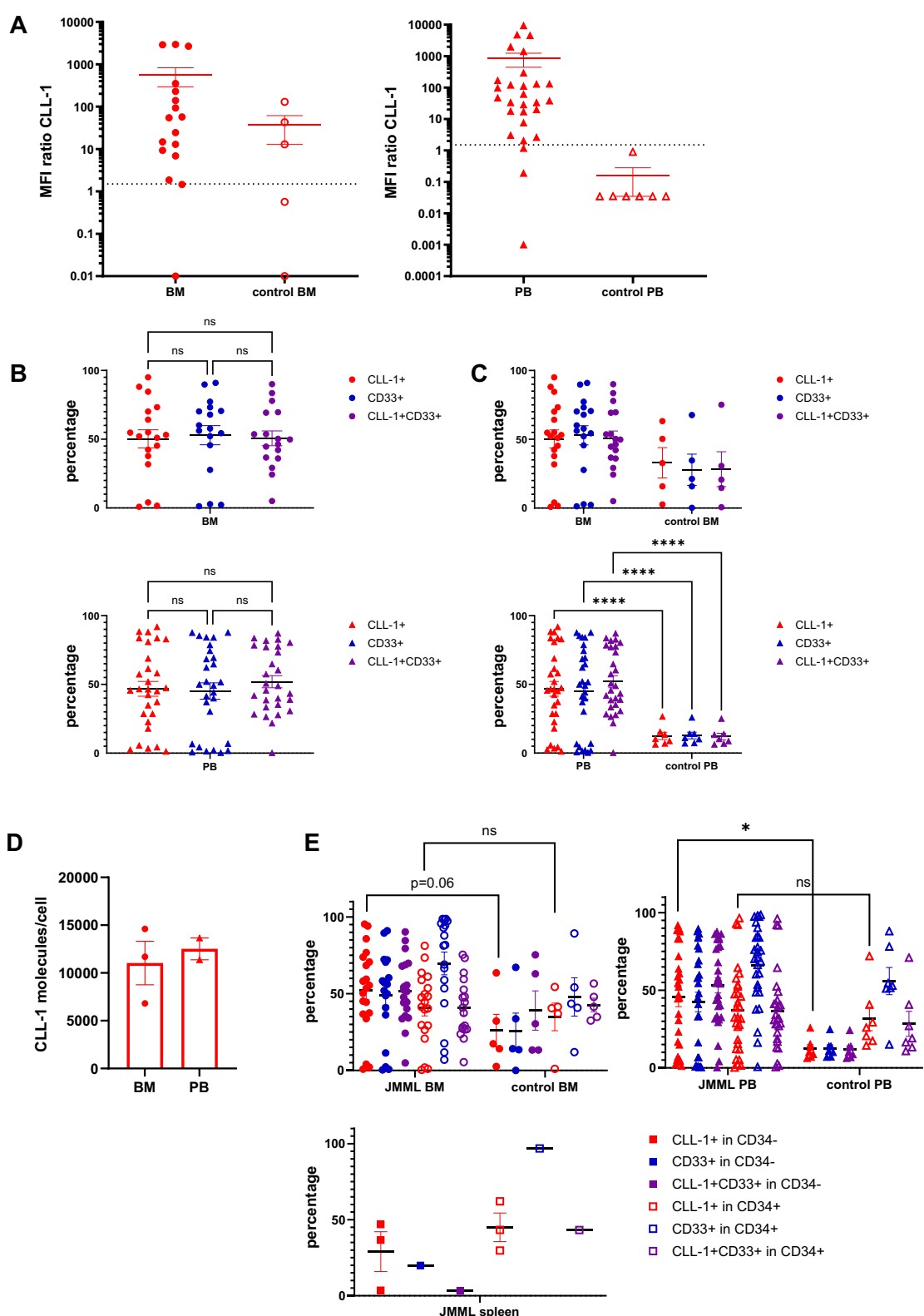

the spleen upon termination (Supplementary Fig. 7). However, *TRAC*-KO reduced TCR-mediated unspecific cytotoxicity of empty CAR and UTD T cells against primary JMML MNCs in vitro (Fig. 5D). We were also interested in assessing readouts of CAR T cell function with and without *TRAC* KO. We thus co-cultured (CAR) T cells with primary JMML MNCs for 24 h. Marked upregulation of CD107a as a marker of degranulation, an important readout of CAR T cell potency, was detected for CLL1CART and CD33CART groups, but not for control empty CAR or UTD T cell negative controls (Supplementary Fig. 8A). We also found that CLL1CART secrete high levels of Th1 cytokines involved in tumor elimination by CAR T cells, as well as Th2 cytokines (Supplementary Fig. 8B). Moreover, CLL1CART showed increases of CD25 and CD69 expression upon JMML exposure in vitro (Supplementary Fig. 9A, B). Lastly, we stained for CD62L and CD45RA to profile

**Fig. 3 | Flow cytometry confirms CLL-1 overexpression on JMML MNCs and LSCs. A** Bulk JMML BM ($n = 17$), healthy donor control BM ($n = 5$), JMML PB ($n = 28$), or healthy donor control PB ($n = 7$) were assessed for the MFI ratio of CLL-1. Positivity for CLL-1 was defined as an MFI ratio > 1.5. **B, C** Bulk JMML BM ($n = 18$ for CLL-1+, $n = 17$ for CD33+ and CLL-1+ CD33+), healthy donor control BM ($n = 5$), JMML PB ($n = 29$ for CLL-1+, $n = 28$ for CD33+ and CLL-1+ CD33+) or healthy donor control PB ($n = 7$) were assessed for percentage of CLL-1, CD33 or double positive cells. Statistical analysis was performed using one-way ANOVA with Tukey's correction for panels just depicting JMML samples, or an unpaired two-tailed t-test to compare protein expression between JMML and healthy control. ****$p < 0.0001$. **D** CLL-1 quantification by flow cytometry on CLL-1+ cells from bulk MNCs from BM ($n = 3$) and PB ($n = 2$). **E** Bulk JMML BM ($n = 20$), healthy donor control BM ($n = 5$), JMML PB ($n = 29$), healthy donor control PB ($n = 7$) or JMML spleen ($n = 3$ for CLL-1, $n = 1$ for CD33) were assessed for percentage of CLL-1, CD33 or double positive cells by flow cytometry. Statistical analysis was performed using an unpaired two-tailed t-test. p-values: CLL-1+ in CD34− JMML vs. control BM = 0.066; CLL-1+ in CD34+ JMML vs. control BM = 0.6110; CLL-1+ in CD34− JMML vs. control PB = 0.0111; CLL-1+ in CD34+ JMML vs. control PB = 0.5958.

CAR T cell memory/stemness markers. CLL1CART had a similarly high combined fraction of naïve (Tn) and central memory T (Tcm) cells compared to control empty CAR and UTD T cell groups (Supplementary Fig. 9C). Exhaustion markers revealed only modest differences between CLL1CART and CD33CART groups (Supplementary Fig. 9D).

## Optimized TRAC-KO CLL1CART immunotherapy inhibit JMML proliferation in vivo

The observed TCR-mediated alloreactivity also occurred in initial in vivo PDX trials (Supplementary Fig. 10A, B). After demonstrating reduced TCR-mediated alloreactivity of our optimized *TRAC*-KO CLL1CART in vitro, we then assessed the in vivo activity of these in two JMML PDX models comprised of different Ras pathway mutations (Supplementary Data 9): First, we tested CLL1CART in an *NRAS*-mutant JMML PDX model in a termination study where all mice were sacrificed at the same timepoint (HM5896, HM = JMML sample identifier; Fig. 6A and Supplementary Fig. 11A). PDX mice were treated four weeks after xenotransplantation. A significant reduction in BM cellularity (Supplementary Fig. 11B), leukemia burden in PB (Fig. 6B) and leukemia (Fig. 6C), as well as LSC (CD34+CD38− and CD34+CD38+ cells) burden was observed in end-study organs of CLL1CART-treated mice compared to negative controls (Fig. 6D–F and Supplementary Fig. 11C) after sequential treatment with CLL1CART. We then validated our findings in a second *NF1*-mutant JMML PDX model (HM2130; Fig. 7A Supplementary Figs. 12 and 13A) treated with a single higher initial dose of *TRAC*-KO CLL1CART and similarly observed reduction of human JMML cells in murine PB and end-study harvested tissues (Fig. 7B, C and Supplementary Fig. 12) by staining for CD14, CD33 and CLL-1 (Supplementary Fig. 13B), which similar to the HM5896 PDX (Supplementary Fig. 11C) represent the majority of human engrafted JMML cells. We did not observe differences in (normalized) spleen length or weight (Supplementary Fig. 13C, D) or BM cellularity (Supplementary Fig. 13E). Importantly, the LSC compartment was also significantly diminished in the CLL1CART-treated mice (Fig. 7D–F and Supplementary Fig. 13F) in all three assessed tissues. Survival analyses of additional HM2130 JMML PDX mice (Fig. 7G and Supplementary Fig. 13G) treated with repeated doses of CLL1CART led to a significant reduction of human JMML burden in murine PB (Fig. 7H and Supplementary Fig. 13H, I) and prolonged animal survival of CLL1CART-treated mice compared to UTD-treated mice (Fig. 7I). A trend towards lower leukemia burden was observed in CLL1CART versus UTD and PBS upon humane endpoint (Supplementary Fig. 14A). We did not observe significant changes in CLL-1 expression in the survival study or the two termination studies (Supplementary Fig. 14B–D) to suggest antigen escape.

## CLL1CART effectively reduce JMML LSCs and the ability to perform serial transplantation

In addition to demonstrating depletion of LSCs by CLL1CART in PDX models, we next assessed the engraftment potential of sorted CLL-1+ versus CLL-1− CD34+CD38− cells. Mice in both groups engrafted with a trend towards higher engraftment for the CLL-1+ cells as measured in PB (Supplementary Fig. 15A–C). Upon termination 83 days post-LSC injection, the CLL-1+ group showed a trend towards higher engraftment (Supplementary Fig. 15D).

We next performed secondary transplantation of JMML cells using BM from the aforementioned HM2130 termination study (Fig. 8A). CLL1CART mice had lower tumor burden in PB over time (Fig. 8B and Supplementary Fig. 16A). Upon termination 61 days after BM injection, CLL1CART mice had significantly lower spleen length and weight (Fig. 8C). This corresponded to significantly lower JMML burden in BM, spleen and cardiac blood (Fig. 8D), as well as lower HSPC burden (Fig. 8E−G and Supplementary Fig. 16B−E). Despite likely heterogeneity in CLL-1 expression in LSC-containing compartments, we observed a reduction in serial transplantability of CLL1CART-treated cells in vivo compared to controls.

## Discussion

Novel proteomic and transcriptomic techniques have allowed for comprehensive investigation to identify optimal CAR T cell targets for ALL[51] and AML[52,53], but have not been performed to date for JMML. Using an integrated and comprehensive transcriptomic and "surfaceomic" approach, we generated the largest JMML bulk and scRNAseq data to date and the first mass spectrometry-based proteomic analysis in this disease.

We demonstrated that CLL-1 is an upregulated surface marker on JMML MNCs by bulk RNAseq and flow cytometry. It can also be overexpressed on LSCs[14] detected across multiple modalities, including scRNAseq, mass spectrometry, and flow cytometry, similar to AML[48,54]. CLL-1 was the only potential CAR T cell target in JMML that was overexpressed in both scRNAseq and mass spectrometry versus healthy controls and had a low expression in healthy tissue.

In this study, we identify CLL-1, which has a favorable "on-target off-tumor" toxicity profile, as a promising candidate for cellular therapy targeting JMML, and we report that CLL1CART have potent in vitro and in vivo effector functions against primary human JMML cells, including LSCs, as evidenced by lower secondary transplantability in vivo of CLL1CART-treated cells compared to appropriate controls. To our knowledge, this is the first report of CAR T cell activity in vivo against JMML.

Our preclinical work credentials a potential new approach for patients with JMML who cannot proceed to standard-of-care allogeneic HSCT or who relapse after HSCT. Current bridging therapies before HSCT include chemotherapy and azacitidine[55], which is the only treatment approved by the Food and Drug Administration for JMML. Recently, the MEK inhibitor trametinib was found to be effective, but not curative, in children with relapsed/refractory JMML who did not respond to upfront therapy[56]. Novel therapies achieving better control prior to HSCT are needed, as patients treated with pre-HSCT therapy achieving molecular remission experienced a trend toward improved outcomes post-HSCT[8]. CLL1CART immunotherapy could provide a new therapeutic approach to address an unmet need for the substantial portion of patients who do not respond to conventional therapies.

Given our observations of potential TCR-mediated alloreactivity of CLL1CART, we created and optimized *TRAC*-KO CLL1CART and demonstrated improved anti-leukemia activity in our preclinical JMML models. We hypothesize that the seemingly large contribution of alloreactivity to the cytotoxic effect of our non-*TRAC*-KO CLL1CART

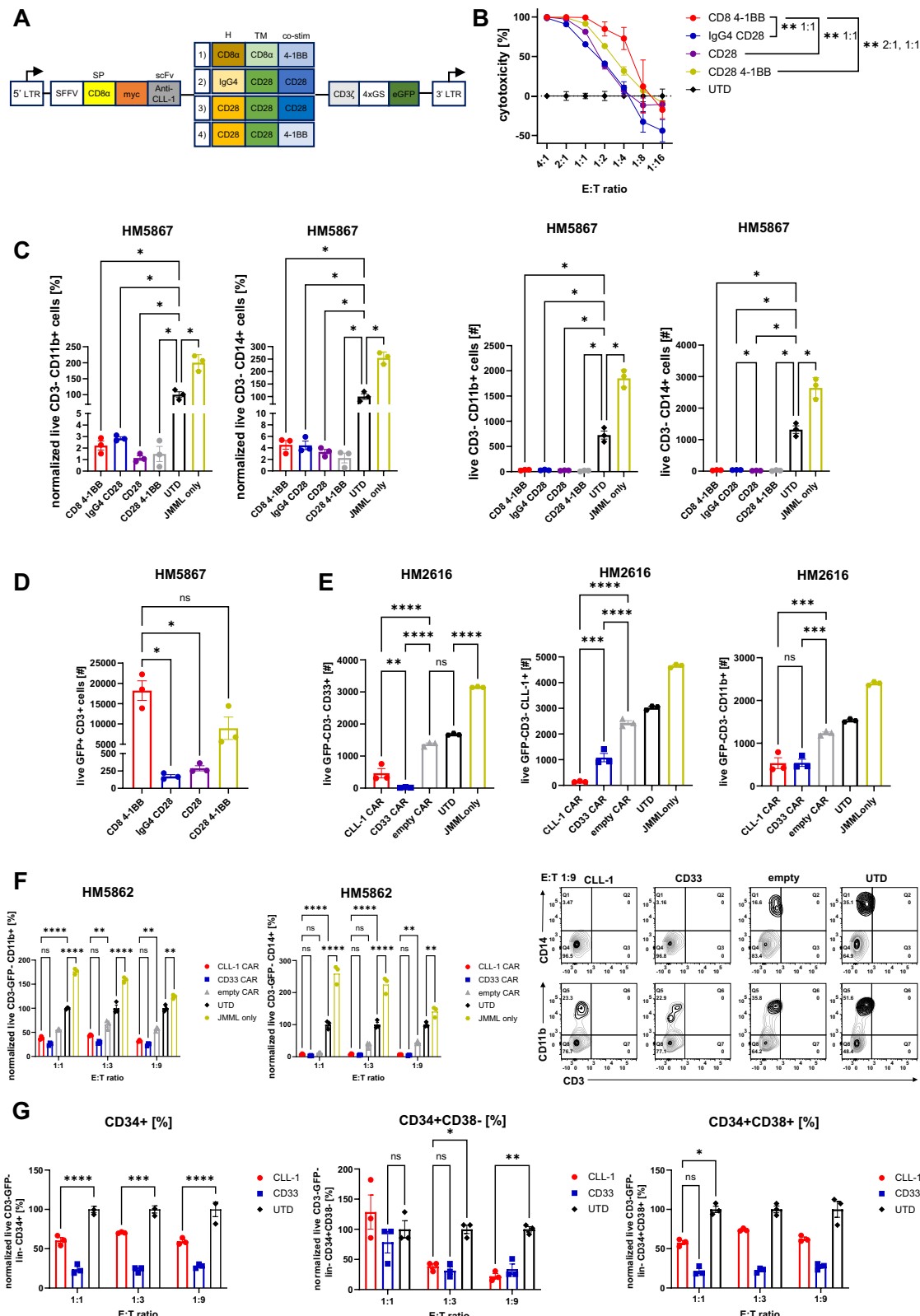

could be due in part to the slower kinetics compared to other more aggressive acute malignancies.

There are several limitations of our study. We observed that CLL-1 was not uniformly overexpressed in all primary patient samples, yet CLL1CART unexpectedly prevented secondary transplantation. How CLL1CART reduced JMML engraftment of HSPCs that lacked CLL-1 expression in these assays is uncertain and requires further study.

However, the decision to treat an individual patient may need to be based on a minimum threshold of CLL-1 expression, as is commonly required in target-based immunotherapy trials. We also observed that even our *TRAC*-KO-optimized CLL1CART did not fully eradicate JMML in vitro and in vivo, and thus posit that optimized adoptive immunotherapy strategies may be necessary to increase CAR T cell potency. However, interventions with CAR T cells in xenograft models of AML[57] are often initiated at earlier time

**Fig. 4 | In vitro cytotoxicity of CLL1CART on cell lines and primary JMML cells.** **A** Schematic of the four second-generation lentiviral CAR vectors with varying H, TM, and co-stim used. **B** In vitro 24-h Luciferase-based cytotoxicity assay comparing the four different CLL1CART against U937. Data are normalized to UTD T cells ($n = 3$ technical replicates). Statistical analysis was performed by mixed-effects model with Geisser−Greenhouse correction and Tukey's multiple comparisons test. *p*-values: CD8 4-1BB vs. IgG4 CD28 E:T 1:1 = 0.0042; CD8 4-1BB vs. CD28 E:T 1:1 = 0.0101; CD8 4-1BB vs. CD28 4-1BB E:T 2:1 = 0.0080; E:T 1:1 = 0.0012. **C** Flow cytometric analysis showing the remaining number of live CD3⁻ CD11b⁺ and CD14⁺ primary JMML (HM5867) cells after 48-h co-culture with the four different CLL1CART ($n = 3$ technical replicates). Statistical analysis was performed by one-way ANOVA with Geisser−Greenhouse correction and Tukey's multiple comparisons test. *p*-values CD11b/CD14: CD8 4-1BB vs. UTD = 0.0278/0.0412; IgG4 CD28 vs. UTD = 0.0310/0.0462; CD28 vs. UTD = 0.0292/0.0432; CD28 4-1BB vs. UTD = 0.0279/0.0432; UTD vs. JMML only = 0.0351/0.0178. **D** Flow cytometric analysis showing the number of live CD3⁺ GFP⁺ CLL1CART after 48-h co-culture with primary JMML cells (HM5867) at E:T 1:1 ($n = 3$ technical replicates). Statistical analysis was performed by one-way ANOVA with Geisser−Greenhouse correction and Tukey's multiple comparisons test. *p*-values: CD8 4-1BB vs. IgG4 CD28 = 0.0432; CD8 4-1BB vs. CD28 = 0.0440; CD8 4-1BB vs. CD28 4-1BB = 0.4435. **E** CLL1CART, positive control CD33CART, negative controls empty CAR or UTD T cells were co-cultured with primary JMML BM cells (HM2616) for 24 h at E:T 1:1. Flow cytometric analysis

showing the remaining number of live non T cell GFP⁻ (to exclude CAR T cells from analysis) CD3⁻ CD33⁺/CLL-1⁺/CD11b⁺ JMML cells ($n = 3$ technical replicates). Statistical analysis was performed by one-way ANOVA with Tukey's multiple comparisons test. *p*-values CD33/CLL-1: CLL-1 CAR vs CD33 CAR = 0.0066/0.0001, ****<0.0001; CD11b: CLL-1 CAR vs empty CAR = 0.0003, CD33 CAR vs empty CAR = 0.0003. **F** Flow cytometric analysis showing the number of remaining live CD3⁻ GFP⁻ CD11b⁺/CD14⁺ JMML cells (HM5862) after co-culture with CLL1CART, CD33CART, or UTD T cells for 24 h at different E:T ($n = 3$ technical triplicates). Statistical analysis was performed by two-way ANOVA with Tukey's multiple comparisons test. *p*-values CD11b: CLL-1 CAR vs. empty CAR E:T 1:3 = 0.0041/E:T 1:9 = 0.0018, UTD vs. JMML only = 0.0023; *p*-values CD14: CLL-1 CAR vs. empty CAR E:T 1:9 = 0.0097, UTD vs. JMML only = 0.0034; ****<0.0001. **G** Flow cytometric analysis showing the number of remaining live CD3⁻ GFP⁻ Lin⁻CD34⁺, CD34⁺CD38⁻ or CD34⁺CD38⁺ JMML HSPCs after co-culture with CLL-1 CAR, CD33 CAR or UTD T cells for 24 h at E:T 1:1 ($n = 3$ technical triplicates). Statistical analysis was performed by two-way ANOVA with Tukey's multiple comparisons test. *p*-values: CD34⁺ CLL-1 vs. UTD E:T 1:3 = 0.0003; CD34⁺CD38⁻ CLL-1 vs. UTD E:T 1:3 = 0.0359/E:T 1:9 = 0.0038; CD34⁺CD38⁺⁻ CLL-1 vs. UTD E:T 1:1 = 0.0365; ****<0.0001. Data in (**B**, **C**, **F**, and **G**) are normalized to UTD T cells. LTR long terminal repeat, SP signal peptide, CD8 4-1BB CD8α H + TM 4-1BB co-stim, IgG4 CD28 IgG4 H + CD28 TM + co-stim, CD28 CD28 H + TM + co-stim, CD28 4-1BB CD28 H + TM + 4-1BB co-stim.

---

points compared to our JMML PDX studies. The higher tumor burden at the time CLL1CART was administered potentially contributed to reduced tumor eradication in our model.

Although we did not directly compare CLL1CART to other targets in vivo, CLL-1 has theoretical advantages over other commonly investigated myeloid targets, including CD33, such as lower expression on normal HSPCs[58], which may reduce prolonged myeloablation[59]. Additionally, in a Phase I/II CD33CART trial in children and young adults with relapsed/refractory AML, only 2/19 infused patients achieved complete remission[60] indicating room for improvement.

Similar to lymphoid malignancies where multiple antigens including CD19, CD20, and CD22 are being targeted, myeloid leukemias will also likely require targeting of multiple antigens either alone or in combination. It is possible that targeting a combination of targets, such as CLL-1, CD33 and CD123 might be most effective in JMML even though CD33 and CD123 did not pass the filters in our bioinformatic pipeline.

Future directions could involve dual antigen-targeting, IL-15 co-expression[61], or combining CLL1CART against JMML with other commonly used treatments in JMML, such as trametinib and azacitidine[2]. Previous reports have shown improvement of CAR T cell function in combination with MEK inhibition[62,63]. Similarly, DNA hypomethylating agents have been shown to reverse DNA methylation programs associated with exhaustion in CAR T cells and thereby improve antileukemic function[64–67] and to upregulate certain surface antigens, which makes them in turn more targetable by CAR T cells[68]. Of note, azacitidine has been shown to be able to target JMML LSCs[69], which further credentials this promising potential therapeutic strategy.

Anti-CLL-1 CAR T cells have already been tested in pediatric[23,24,26] and adult patients[25,27] and have been shown to be safe and effective. Ongoing phase 1 clinical trials in pediatric and adult AML in the United States (NCT04219163, NCT04789408, NCT06017258, NCT06128044, and NCT04789408) may also facilitate the development of CLL1CART immunotherapy for children with JMML, who are currently excluded from these clinical trials. Our results suggest that CLL-1 is a rational target in JMML and support clinical translation and testing of CLL1CART in children with relapsed/refractory JMML.

## Methods
### Bulk and scRNAseq
Bulk: Total RNA was extracted from PB or BM samples using the Qiagen AllPrep kit and was used to prepare mRNA libraries. Target coverage was 40e6 paired end reads per sample.

sc: Samples were multiplexed using the MULTISeq approach on three lanes of the 10× Chromium scRNASeq platform. Target coverage was 2000 cells per sample.

Both: Libraries were prepared for 75-bp strand-specific paired-end sequencing using the ribodepletion v2.0 protocol by the British Columbia Genome Sciences Center (BCGSC, Vancouver, Canada). Libraries were sequenced on the Illumina HiSeq 2000/2500 and aligned to the hg19 (GRCh37-lite) reference genome using BWA v0.5.7 with default parameters, except the addition of "-s" option, and duplicate reads were marked with Picard Tools. Gene-level quantification was performed using the BCGSC-pipeline v1.1 with Ensembl v69 annotations.

### Bulk RNAseq analysis
Raw FASTQ files were obtained directly from the sequencing facility. Preprocessing steps such as trimming adapters and low-quality regions were completed using the Trimmomatic tool (v0.36). All reads were then subsequently aligned to the human reference genome (hg38) using the STAR aligner (v2.5.1b). For quantifying gene-level expression, the 'quantMode' feature of the STAR aligner was employed, with gene annotations sourced from GENCODE p5. Quality control metrics, including counts of uniquely aligned reads, ratios of unique-to-multiple alignments, and exon-to-intron ratios, were assessed using ngsutilsj (v0.3-2180ca6). Further quality control assessments and subsequent analyses were conducted with R. After filtering for samples with >30 million total mapped reads, >3 million non-scaled library reads, >3 million unique mapped reads, and a mitochondrial ratio < 10%, 85 samples were left and analyzed. To assess normalization, density plots were generated. Reads were normalized using the trimmed mean of M-values (TMM) method from the EdgeR package (v3.24.3), with counts transformed as log2(counts per million (CPM) + 1). For identifying statistically significant changes in gene expression, we applied the VOOM function within the limma package to model the mean-variance relationship and generate precision weights. These weights were then incorporated into limma's (v3.38.3) empirical Bayes linear modeling framework to produce statistical measures, including *p*-values, adjusted *p*-values, and log2FC. Additionally, gene set enrichment analysis (GSEA) was performed on the log2FC values, ranked by their significance, using the fgsea package in R (v1.13.5). Control samples are two pediatric PB samples, and five CB HSC samples from GEO (GSE111895). Identification of markers as being present on

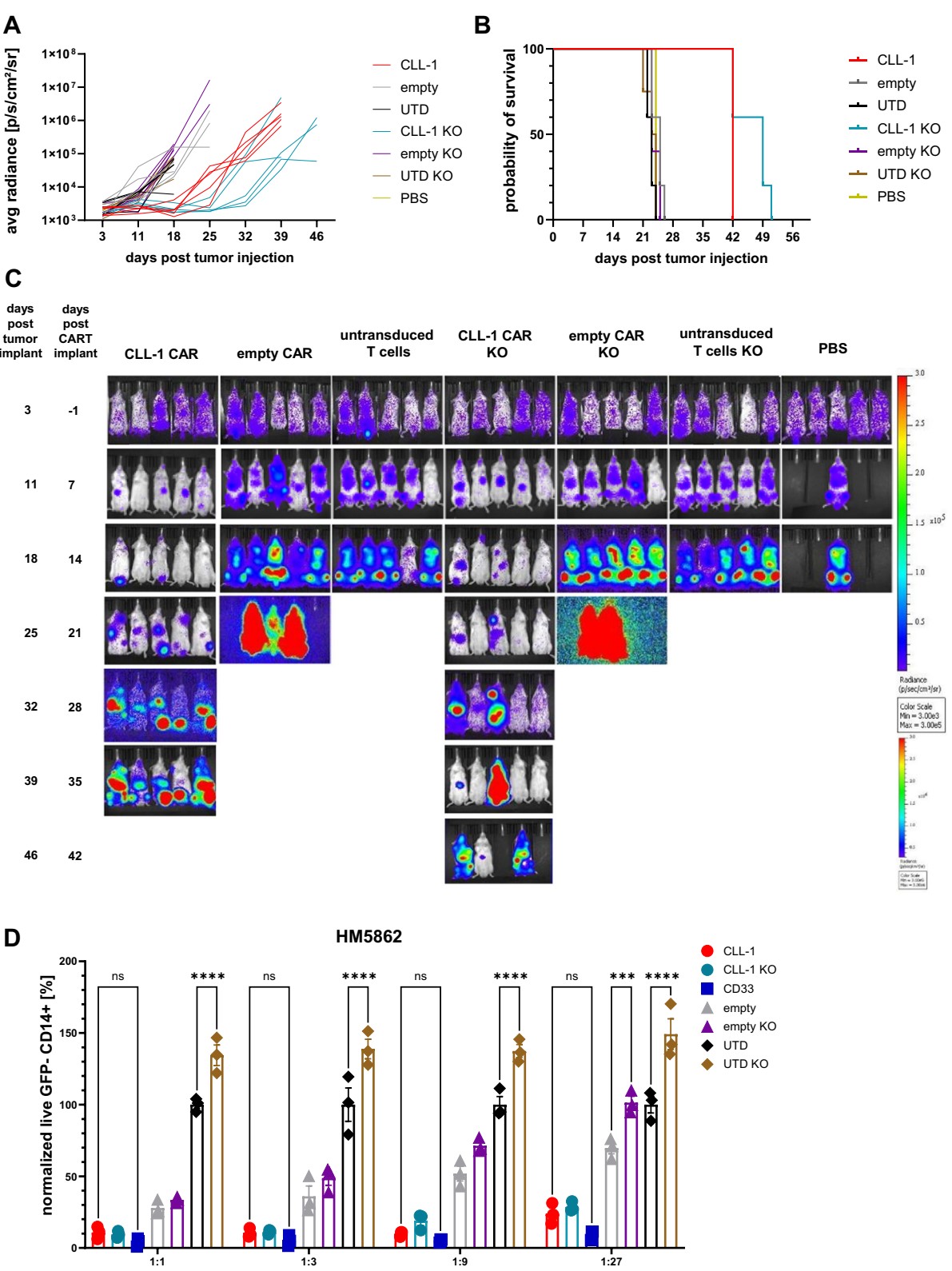

**Fig. 5 | *TRAC*-KO is only relevant to reduce unspecific cytotoxicity in JMML but not the U937 model. A** Quantified bioluminescence imaging of U937 histiocytic lymphoma model in NSG mice treated with regular or *TRAC*-KO CAR or UTD T cells, *n* = 5/arm except *n* = 1 for PBS. **B** Survival curve for the same study as in (**A**). **C** Bioluminescence images for the same study as in (**A**). Note increased scale for bioluminescence measurements at later timepoints. **D** Flow cytometric analysis showing the number of remaining live GFP⁻ CD14⁺ JMML cells (normalized to UTD T cells) after co-culture with CAR or control T cells for 24 h at different E:T (*n* = 3 technical triplicates). Statistical analysis was performed by two-way ANOVA with Tukey's multiple comparisons test. *p*-values: empty vs. empty KO E:T 1:27 = 0.0001, ****<0.0001. avg radiance average radiance.

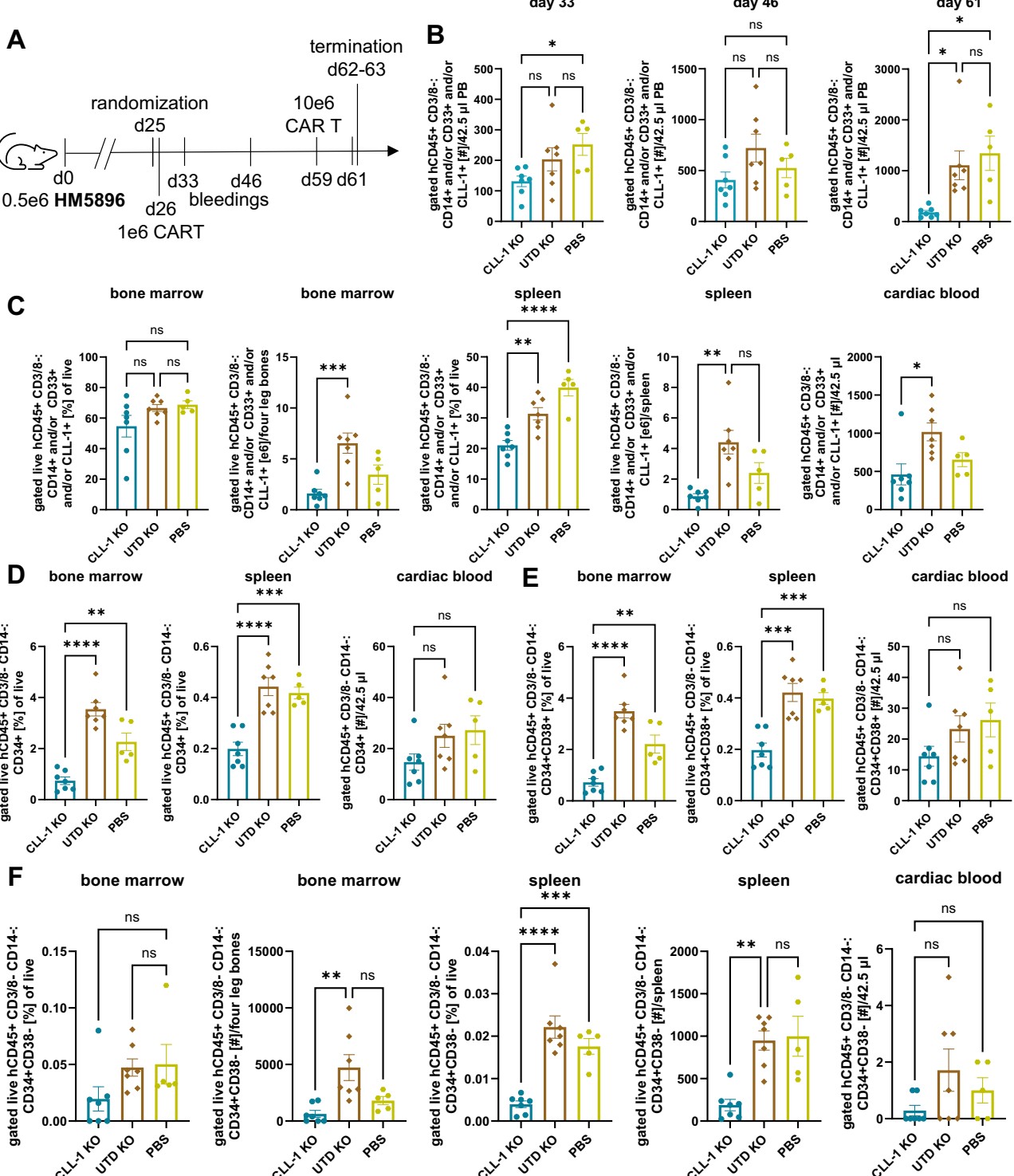

**Fig. 6 | CLL1CART with *TRAC*-KO (CLL-1 KO) is effective against the HM5896 PDX model. A** Schematic of the murine trial design with injection of 1e6 and later 10e6 CAR or UTD T cells. **B** Flow cytometric assessment of JMML burden in PB over time. *p*-values: d33 CLL-1 KO vs. PBS = 0.0452; d61 CLL-1 KO vs. UTD KO = 0.0268, CLL-1 KO vs. PBS = 0.0114. **C** Flow cytometric assessment of leukemia burden in different tissues upon termination. *p*-values: BM [e6] CLL-1 KO vs. UTD KO = 0.0009; spleen [%] CLL-1 KO vs. UTD KO = 0.0046; spleen [e6] CLL-1 KO vs. UTD KO = 0.0011; cardiac blood CLL-1 KO vs. UTD KO = 0.0104; ****<0.0001. **D** Flow cytometric assessment of stem cell amount by staining for CD34 in different tissues

upon termination. *p*-values: BM/spleen CLL-1 KO vs. PBS = 0.0019/0.0004; ****<0.0001. **E, F** Flow cytometric analysis subsetting cells from (**D**) into CD34⁺CD38⁺ cells (**E**) and CD34⁺CD38⁻ cells (**F**). *p*-values **E** BM CLL-1 KO vs. PBS = 0.0023; spleen CLL-1 KO vs. UTD KO = 0.0001, CLL-1 KO vs. PBS = 0.0010; **F** BM [#] CLL-1 KO vs. UTD KO = 0.0035; spleen [%] CLL-1 KO vs. PBS = 0.0006; spleen [#] CLL-1 KO vs. UTD KO = 0.0019; ****<0.0001. *n* = 7 mice for CLL-1 KO and UTD KO (CLL-1 or UTD with *TRAC*-KO), *n* = 5 for PBS. Statistical analysis for (**A–F**) was performed by one-way ANOVA with Tukey's multiple comparisons test. d day.

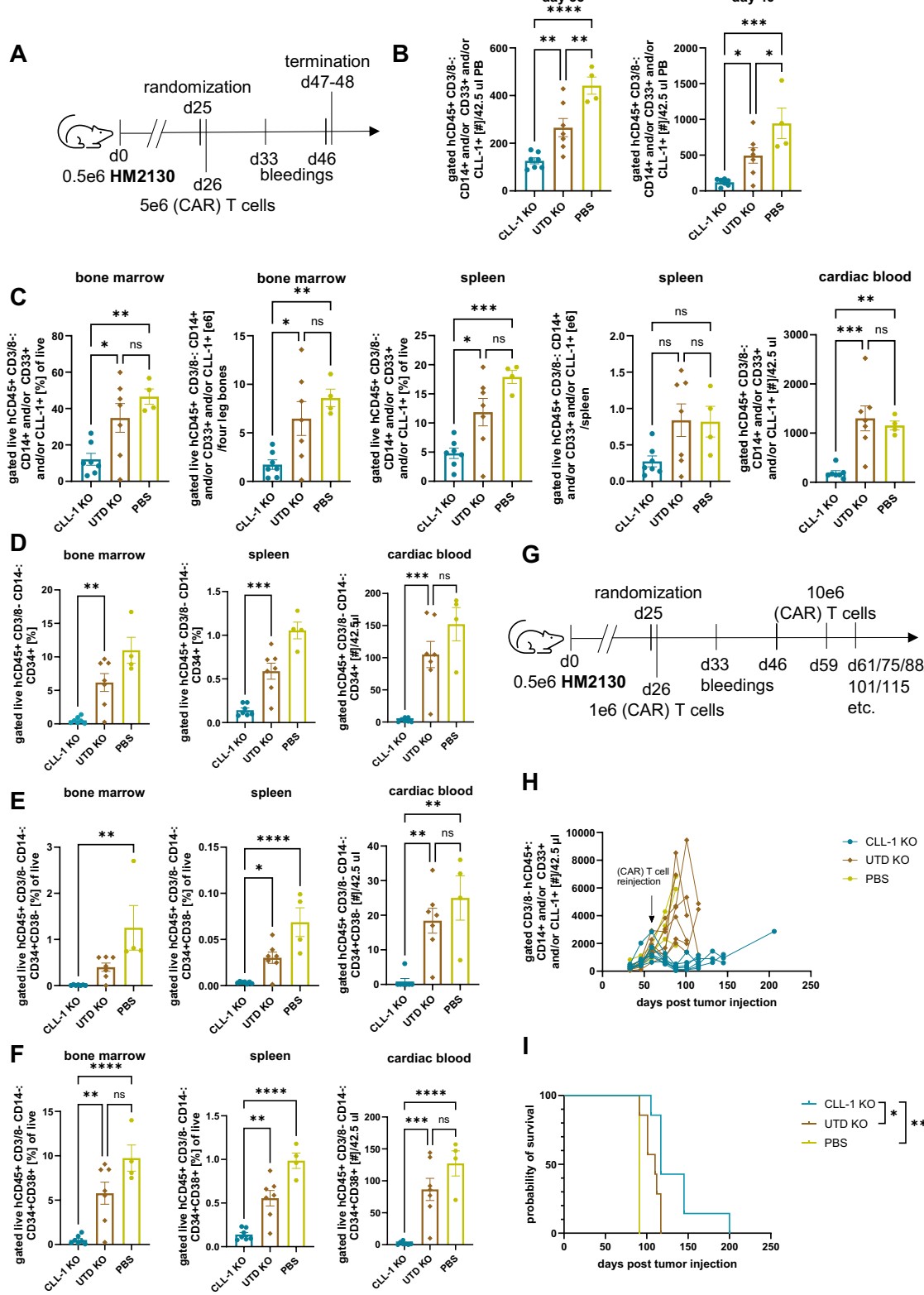

scRNAseq analysis

the cell surface was based on annotation in at least one of the following resources: the Human Protein Atlas, Bausch-Fluck et al.[28], and Hu et al.[29]. For the comparison with GTEx (downloaded from the GTEx portal: GTEx_Analysis_2017-06-05_v8_RNASeQCv1.1.9_gene_tpm), the data was downloaded from Xena UCSC, which utilizes the same TOIL RNAseq pipeline as we used. Figures were generated in R (v 4.1.2).

## scRNAseq analysis

The 22 JMML samples and 1 healthy donor pediatric BM sample were run on the 10× platform. Another four healthy donor pediatric BM samples were obtained from Chen et al. [70]. Quality control and pre-processing were based on Seurat's recommendations. Briefly, we filtered out cells with fewer than 200 or greater than 4000 unique feature counts, less than 1000 total RNA molecules, and

**Fig. 7 | CLL1CART with *TRAC*-KO is effective against the HM2130 PDX model.**
**A** Schematic of the murine trial design with injection of 5e6 CAR or UTD T cells.
**B** Flow cytometric assessment of JMML burden in PB over time. *p*-values d33/d46:
CLL-1 KO vs. UTD KO = 0.0087/0.0451, CLL-1 KO vs. PBS = ****/0.0005, UTD KO vs.
PBS = 0.0050/0.0394; ****<0.0001. **C** Flow cytometric assessment of leukemia bur-
den in different tissues upon termination 47 days after (CAR) T cell injection.
*p*-values CLL-1 KO vs. UTD KO BM [%]/BM [e6]/spleen [%]/cardiac blood = 0.0293/
0.0323/0.0204/0.0008; CLL-1 KO vs. PBS BM [%]/BM [e6]/spleen [%]/cardiac
blood = 0.0055/0.0087/0.0006/0.0085. **D** Flow cytometric assessment of LSC
amount by staining for CD34 in different tissues upon termination. *p*-values CLL-1
KO vs. UTD KO BM/spleen/cardiac blood = 0.0055/0.0009/0.0009. **E, F** Flow

cytometric analysis subsetting cells from (**D**) into CD34+CD38− (**E**) and CD34+CD38+
cells (**F**). *p*-values (**E**) CLL-1 KO vs. UTD KO spleen/cardiac blood = 0.0271/0.0039,
CLL-1 KO vs. PBS BM/spleen/cardiac blood = 0.0016/****/0.0010; *p*-values (**F**) CLL-1
KO vs. UTD KO BM/spleen/cardiac blood = 0.0033/0.0011/0.0008; ****<0.0001.
**G** Schematic of murine trial design with injection of 1e6 and later 10e6 CAR or UTD
T cells. *n* = 7 mice for CLL-1 KO and UTD KO, *n* = 3 for PBS. **H** Flow cytometric
assessment of JMML burden for the same study as in (**G**) in PB over time. **I** Survival
curve for the study in (**G**). Statistical analysis was performed by the Log-rank
(Mantel−Cox) test. *p*-values: CLL-1 KO vs. UTD KO = 0.0161, CLL-1 KO vs. PBS =
0.0027. *n* = 7 mice for CLL-1 KO and UTD KO, *n* = 4 for PBS. Statistical analysis for
(**B**−**F**) was performed by one-way ANOVA with Tukey's multiple comparisons test.

mitochondria levels >15% (Supplementary Fig. 17). Normalization
and variance stabilization were performed using the SCTransform
function regressing out mitochondria levels. Final integration and
batch correction were performed using Harmony[71]. Dimensionality
reduction with UMAP was then used to identify cell clusters, and cell
type was subsequently identified algorithmically using ScType
(Supplementary Data 14)[72]. To further refine our characterization of
these clusters, we referenced the immunophenotypic markers for
HSPCs as described by Louka et al.[14] and employed the scGate
algorithm[73]. This is an automated signature scoring method allowing
us to construct a synthetic gating model for each hematopoietic
lineage for individual cells, enabling a more granular analysis cell-by-
cell instead of labeling the entire clusters. Differentially expressed
gene (DEG) analysis was then performed between the HSC clusters of
JMML vs. healthy controls. Identification of markers as being present
on the cell surface was based on the Human Protein Atlas and Bausch-
Fluck et al.[28]. We further subset DEG genes based on normal tissue
acquired from GTEx TPM counts (see above). Gene names are based
on HUGO-approved symbols. All analyses, including figures, were
generated in R (v 4.1.2).

### Patient samples and cell lines
Primary JMML and healthy donor control samples were obtained after
written, informed consent at the Benioff Children's Hospital at the
University of California, San Francisco under a locally approved insti-
tutional review board research protocol and in accordance with the
Declaration of Helsinki. Consent to this study allowed for sharing
anonymized clinical data. This complied with all relevant ethical reg-
ulations. The samples have limited availability due to the nature of the
biological materials. Ficoll density centrifugation-purified MNCs from
BM or PB were used for the experiments. CD34 selection for flow
cytometric experiments was performed using a CD34 microbead kit
(EasySep Human CD34 Pos Selection Kit II) (StemCell Technologies,
17856) according to the manufacturer's instructions. Purity of CD34+
cells was higher than 70% as determined by post-enrichment flow
cytometric analysis. Primary cells were cryopreserved in 90% fetal
bovine serum (FBS) and 10% dimethylsulfoxide. Primary JMML sample
data were collected and managed using REDCap electronic data cap-
ture tools hosted at UCSF[74,75]. There is a known sex imbalance in JMML
with a 2.5:1 male:female ratio. All samples available were included
irrespective of sex. Sex was self-reported by families and was not
considered in the study design due to the rarity of the disease and the
small sample size of the study.

Jurkat, THP-1, and U937 cells were cultured in RPMI-1640 medium,
10% FBS, and 100 U/mL penicillin-streptomycin. These cell lines were
authenticated by short tandem repeat (STR) analysis. K562 was cultured
in IMDM medium, 10% FBS, and 100 U/mL penicillin-streptomycin. Jur-
kat cells were tested for mycoplasma contamination using the Lookout
Mycoplasma PCR Detection kit (Sigma-Aldrich, MP0035-1KT).

Human embryonic kidney 293 T cells and NIH-3T3 cells were
cultured in DMEM medium supplemented with 10% FBS, 2 mM L-glu-
tamine (Glutamax I from Life Technologies Corporation, 35050061),

10 mM HEPES (Life Technologies Corporation, 15630080), and 1 mM
sodium pyruvate.

All cells were cultured at 37 °C in a humidified incubator with
5% $CO_2$.

### Flow cytometry
Cells were labeled at 4 °C after blockade of Fc receptors. Murine blood
and BM were treated with ACK buffer or BD Lysing solution for red
blood cell lysis. Antibodies are listed in Supplementary Data 10−12.
Brilliant Stain Buffer Plus (BD Biosciences, 566385) was added when
using multiple Brilliant Violet antibodies. Antibodies were titrated
using the staining index. Compensation was performed using BD
CompBeads (BD Biosciences, 51-90-9001229), ArC reactive beads (Life
Technologies, 2480653), and GFP BrightComp Beads (Life Technolo-
gies, A10514). Data were acquired on LSRII (BD Biosciences) or Attune
Nxt cytometers (Thermo Fisher Scientific) and analyzed in FlowJo (BD
Biosciences). The buffer used contained D-PBS, 2% FBS, and ethyle-
nediaminetetraacetic acid (EDTA).

For CAR T cell in vitro phenotyping, CAR T cells were cultured
with primary JMML MNCs at E:T 1:1 and stained after 24 h for activation,
exhaustion, and memory phenotype.

Flow cytometric analysis for CLL-1 expression included the fol-
lowing gates in subsequent order: cells of interest in FSC-A/SSC-A gate,
doublet exclusion in FSC-A/FSC-H gate, dead cell exclusion in live dead
(LD) plot, selection of hCD45+ cells, and then either CLL-1 vs. CD33
expression or CD34 vs. CD38 expression. From the latter, CLL-1 vs.
CD33 expression was also determined in different subpopulations of
CD34+/− and CD38+/− cells.

Representative gating strategies are shown in Supplementary
Figs. 18−28.

### Vector construction and cloning
The four lentiviral CAR backbones vary in their H, TM, and co-stim. All
include a CD8α signal peptide and CD3ζ signaling domain. The pre-
viously published CLL-1 single chain variable fragment (scFv)
sequence[22,61] was kindly provided by Prof. Malcolm K. Brenner, Baylor
College of Medicine, Houston, USA. It was cloned into the different
CAR backbone plasmids using Gibson Assembly (NEB, E2611S). DNA
sequencing was performed to confirm the accuracy of the vector. CAR
expression vectors utilized an eGFP marker for identification of CAR+
cells. Final constructs were then expressed in NEB Stable Competent
*E. coli* (High Efficiency) (NEB, C3040H). DNA was isolated using
NucleoBond Xtra Midi Kit for plasmid DNA (Macherey-Nagel,
740410.50) or QiaPrep Spin Miniprep Kit (Qiagen, 27104).

### Structural modeling
The X-ray crystal structure of CLL-1 (amino acids 132-253, PDB 8JAH)
was docked against the predicted structure of the M26 scFv CAR using
HADDOCK and visualized in UCSF ChimeraX. The CAR structure was
generated using OmegaFold Collab (32 cycles). In the HADDOCK web
application, active residues were as follows (CLL-1: 101-256, M26 heavy
chain CDR1-3, M26 light chain CDR1-3), and 400 structures were used

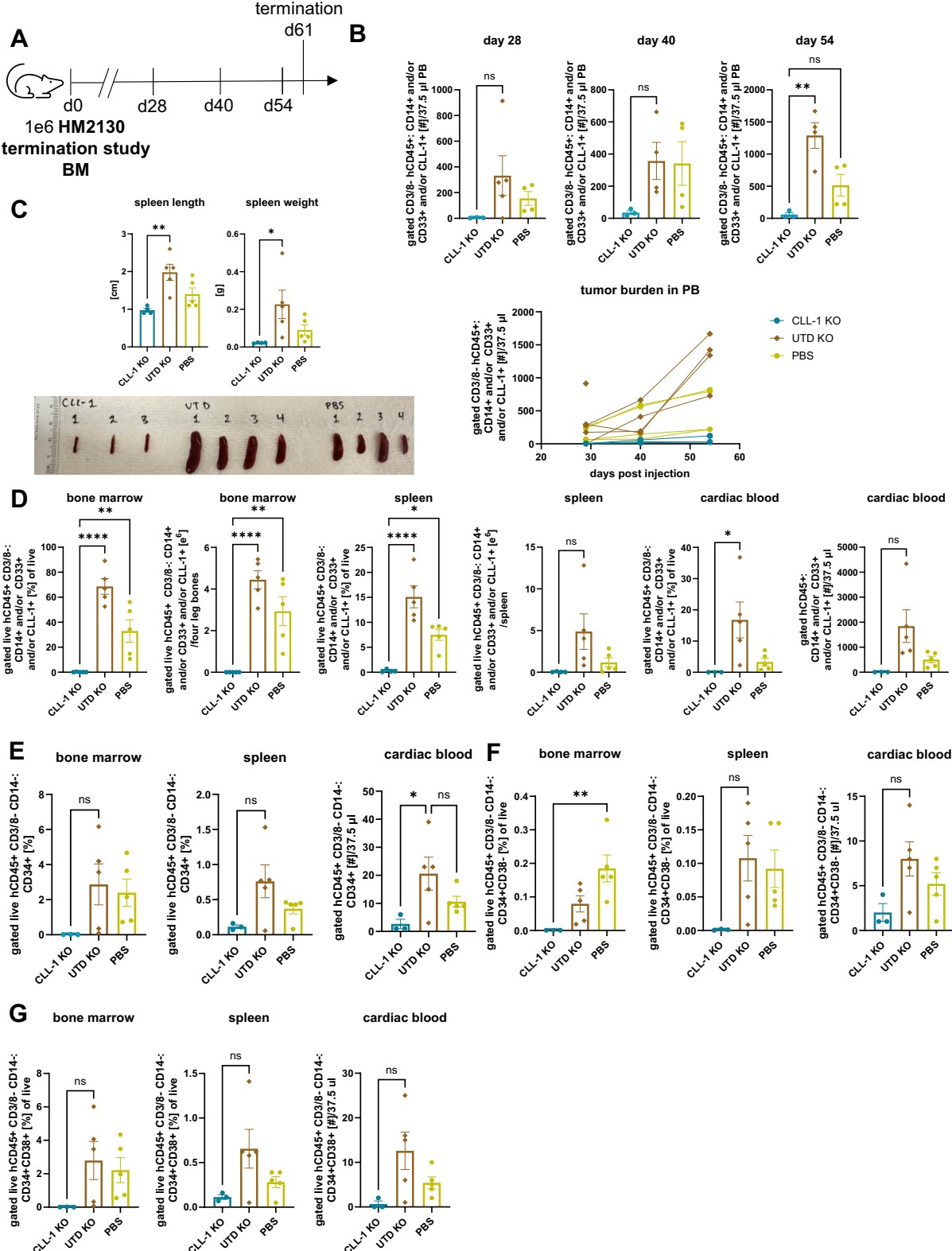

**Fig. 8 | CLL1CART eradicate LSCs and reduce serial transplantability.**
**A** Schematic of the murine trial design with injection of 1e6 BM cells from the
CLL1CART, UTD, or PBS group of the HM2130 termination study. **B** Flow cyto-
metric assessment of JMML burden in PB over time. *p*-values: d54 CLL-1 KO vs. UTD
KO = 0.0027. **C** Spleen length and spleen weight upon termination. *p*-values: CLL-1
KO vs. UTD KO spleen length/spleen weight = 0.0049/0.0407. **D** Flow cytometric
assessment of leukemia burden in different tissues upon termination 61 days after

(CAR) T cell injection. *p*-values: CLL-1 KO vs. PBS BM [%]/BM [e6]/spleen
[%] = 0.0084/0.0024/0.0194; CLL-1 KO vs UTD KO cardiac blood [%] = 0.0480.
**E** Flow cytometric assessment of CD34+ cells in different tissues upon termination.
*p*-value: CLL-1 KO vs. UTD KO cardiac blood=0.0470. **F, G** Flow cytometric analysis
subsetting cells from (**E**) into CD34⁺CD38⁻ (**F**) and CD34⁺CD38⁺ (**G**) cells. *p*-value:
**F** BM CLL-1 KO vs. PBS = 0.0087. *n* = 5 mice/group. Statistical analysis for (**B**–**G**) was
performed by one-way ANOVA with Tukey's multiple comparisons test.

for final refinement and clustering. The lowest HADDOCK energy score was selected for visualization and analysis of potential contacts (19 M26 residues less than 3.5 Angstroms between CLL-1 and M26) using UCSF ChimeraX.

## Virus generation

For transient viral vector production, 293 T cells were transfected using Mirus TransIT-LT1 (Mirus Bio, MIR 2305) with CAR constructs. They were combined with plasmids encoding gag/pol and the VSVG envelope. ViralBoost Reagent (ALSTEM, VB100) was added to enhance viral production. Lentiviral supernatant was filtered after a maximum of 72 h through a PES 0.45 μm filter (Fisher Scientific, SLHPR33RS). It was either freshly used or concentrated by Lenti-X concentrator (Takara Bio, 631232), immediately frozen in dry ice, and stored at −80 °C for further usage. The NIH-3T3 cell line was used for determining the viral titer.

## CAR activation testing in Jurkat T cells

For the transduction of Jurkat T cells, 24-well plates were coated with Retronectin (Takara) overnight at 4 °C. Lentiviral supernatant was added, and plates were spin-oculated (1500 rpm, 90 min, 32 °C). Subsequently, lentiviral supernatant was removed, and Jurkat T cells were added with 5 μg/mL polybrene. CAR-expressing Jurkat T cells were incubated with target cells at a 1:1 ratio for 24 h, then assessed for activation by staining for CD69 upregulation by flow cytometry (Supplementary Fig. 29).

## Generation of CAR T cells

Primary human CD8 and CD4 T cells were isolated from leukapheresis products of anonymous healthy blood donors (StemCell Technologies). CD8$^+$ and CD4$^+$ T cell populations were isolated separately using EasySep Human CD8 T cell Isolation Kit (StemCell Technologies, 17953) or EasySep Human CD4 T cell Isolation Kit (StemCell Technologies, 17952). Purity post selection was assessed by flow cytometry and was >85% for CD8 and >90% for CD4 T cells. CD8 T cells were either cultured alone or mixed at a 1:1 ratio with CD4 after thawing for experiments. T cells were cultured in CTS OpTmizer T Cell Expansion SFM (Thermo Fisher Scientific, A1048501) supplemented with 10% human heat-inactivated AB serum (Valley Medical, HP1022HI or BIOIVT, HUMANABSRMP-HI-1) and were passaged every two days. For expansion, T cells were stimulated with CD3/CD28 dynabeads (Life Technologies, 11131D) according to the manufacturer's instructions (20 μL/1e6 T cells) for five days and stimulated with the following human cytokines: interleukin-7 (IL-7, 10 ng/mL) (PeproTech, 200-07-50UG) and interleukin-15 (IL-15, 10 ng/mL) (PeproTech, 200-15-50UG). Transduction with CAR lentivirus was performed one day after the start of bead stimulation by adding thawed lentivirus. After the removal of activation beads, T cells were MACS-enriched for some experiments using anti-myc and anti-biotin MicroBeads (Miltenyi Biotec, 130-090-485). CAR T cell numbers used in all experiments were adjusted based on a percentage of CAR expression as determined by GFP positivity. Representative transduction efficiencies are shown in Supplementary Fig. 27B, C.

## TRAC-KO

Cas9 protein (40 μM; Macrolab, University of California, Berkeley) and sgRNA (80 μM; Synthego Corporation) were mixed in 1:2 molar ratio and incubated at 37 °C for 15 min. 2e6 of primary T cells were spun down, resuspended in 20 μL of P3 nucleofection solution (P3 Primary Cell 96-well Nucleofector Kit from Lonza, V4SP-3096) plus ribonucleoprotein (RNP) complex, and nucleofected using EH-115 nucleofection program in a Lonza 4D-Nucleofector. Eighty microliters of warm CTS OpTmizer T Cell Expansion SFM was plated into each well and incubated at 37 °C for 15 min, then transferred to an appropriately sized plate or flask supplemented with IL-7 and IL-15 to recover for 72 h.

Five days post-electroporation, T cells with successful *TRAC*-KO were enriched by using a human CD3 MicroBeads kit (Miltenyi Biotec, 130-050-101). The sgRNA sequence was CAGGGTTCTGGATATCTGT[50].

## Cytotoxicity assay

Effector CAR T cells and target cells (cell line transduced, single-cell sorted, and expanded to stably express firefly Luciferase-mCherry) were co-cultured in tumor cell media at indicated ratios for 24 h. Cytotoxicity was assessed by bioluminescence after the addition of D-Luciferin (Gold Biotechnology, LUCK-1G) using a Tecan infinite m200 pro. Percent viable cells were normalized to the bioluminescence of target cells incubated with UTD T cells at the same E:T ratios.

For assays with primary JMML cells the culture medium consisted of IMDM with 10% FBS, 1-Thioglycerol, ascorbic acid (50 μg/ml), L-glutamine (2 mM), SCF, Flt3l, TPO and IL-3 (each 100 ng/mL) and GM-CSF (10 ng/mL).

## Incucyte live-cell killing assays

CAR T cells or UTD T cells were co-cultured with tumor cells at different E:T ratios. For A375 cells, 1e4 cells/well were seeded in a flat-bottom clear 96-well plate. 200 μL was used per well. In the case of A375 cells, 100 μL of OpTmizer media (same as above) with the addition of 2% glucose solution and 100 μL of RPMI-based tumor medium were used. A375 cells were engineered to express nuclear mKate. Data were analyzed using the Incucyte Live-Cell Analysis system (Sartorius). Co-culture plates were incubated for multiple days in the Incucyte, and images were collected every three to four h. Data were normalized to the initial time point and plotted over time.

## Degranulation assay

CAR T cells were co-cultured with tumor at E:T ratio of 1:1 for 6 h with CD107a-APC antibody (BD Biosciences, 560664) and Protein Transport Inhibitor (containing Monensin) GolgiStop (BD Biosciences, 554724). Levels of CD107a were then measured with a flow cytometer as a readout of degranulation.

## CAR T cytokine release assays

Various CAR T cells or UTD T cells were co-cultured with primary JMML target cells at a 1:1 E:T ratio. After 24 h, cells were centrifuged, supernatant was isolated, and then snap frozen in liquid nitrogen. Cytokine samples were diluted 1:1 in CTS OpTmizer T Cell Expansion SFM without cytokines. Eve Technologies used Luminex xMAP technology for multiplexed quantification of human cytokines, chemokines, and growth factors. The multiplexing analysis was performed using the Luminex™ 200 system (Luminex) by Eve Technologies Corp. Fourteen markers were simultaneously measured in the samples using Eve Technologies' Human High Sensitivity 14-Plex Discovery Assay® (MilliporeSigma) according to the manufacturer's protocol. The 14-plex consisted of GM-CSF, IFNγ, IL-1β, IL-2, IL-4, IL-5, IL-6, IL-8, IL-10, IL-12p70, IL-13, IL-17A, IL-23, and TNF-α. Assay sensitivities of these markers range from 0.11 pg/mL to 3.25 pg/mL for the 14-plex. Individual analyte sensitivity values are available in the MilliporeSigma MILLIPLEX® MAP protocol.

## Cell surface protein labeling

CD34$^+$ cells were selected with a human CD34 MicroBead Kit UltraPure (Miltenyi Biotec, 130-100-453). We used a previously published small-scale cell surface biotinylation to label N-acetylglucosamine and sialic acid glycans on the surface[76] of CD34$^+$ cells from JMML PB and BM or CD34$^+$ cells from healthy donor G-CSF stimulated leukapheresis products. Briefly, cells were resuspended in PBS containing 0.04 mg/mL of wheat germ agglutinin-horseradish peroxidase (WGA-HRP; Vector Biolaboratories, PL-1026). Biotin tyramide (Sigma-Aldrich, SML2135)

was added at a final concentration of 500 μM and mixed thoroughly before the addition of 1 mM H2O2 (Sigma-Aldrich, H1009). Cells were labeled in a heated shaker (1500 rpm) at 37 °C for 2 min before being quenched with 9.9 mg/mL (+)-Sodium ʟ-ascorbate (Sigma-Aldrich, A4034)/5 mM sodium pyruvate. Cells were washed twice in quench buffer. The pellet was snap-frozen in liquid nitrogen and stored at −80 °C.

### Cell lysis, cell surface protein enrichment, and peptide digestion

Frozen cell pellets were thawed on ice, then 500 μL of 2× radio-immunoprecipitation assay (RIPA) buffer (Millipore Sigma, 20-188) with 1× Halt protease inhibitors (Thermo Fisher Scientific, 1861280) and 1.25 mM EDTA (Invitrogen, 15575-038) were added. Cells were lysed by sonication, and the lysates were centrifuged at 17,000×$g$ at 4 °C for 10 min to obtain the clarified lysate containing the cell surface proteins. Clarified lysate was mixed with 100 μL of NeutrAvidin agarose resin (Thermo Fisher Scientific, 29204) and incubated at 4 °C for 2 h on an end-over-end rotator. NeutrAvidin beads with captured biotinylated surface proteins were washed in columns attached to a vacuum manifold to remove unbound proteins using 5 mL of 1× RIPA + 1 mM EDTA, followed by 5 mL of PBS + 1 M NaCl, and lastly 5 mL of 50 mM ammonium bicarbonate (ABC) + 2 M urea (VWR, M123) buffer. Washed beads were resuspended in digestion buffer (from PreOmics, P.O.00027) with added trypsin protease to perform simultaneous disulfide reduction, alkylation of cysteine residues, and on-bead peptide digestion for 90 min at 37 °C and 700 rpm shaking. Then, the peptide mixture was desalted, eluted, and completely dried in a vacuum concentrator (Labconco, 7810010). Dried peptides were resuspended in 2% acetonitrile (ACN) and 0.1% formic acid. Peptide concentration was measured using Protein205A on a NanoDrop (Thermo Fisher Scientific), and the peptide concentration was adjusted to 0.1 μg/μL for mass spectrometry runs.

### Liquid chromatography/mass spectrometry and data analysis

A nanoElute was attached in line to a timsTOF Pro equipped with a CaptiveSpray Source (Bruker Daltonics). Chromatography was conducted at 50 °C through a 25 cm reversed-phase C18 column (PepSep) at a constant flow rate of 0.5 μL min⁻¹. Mobile phase A was 98/2/0.1% water/MeCN/formic acid ($v/v/v$) and phase B was MeCN with 0.1% formic acid ($v/v$). During a 105 min method, peptides were separated by a 7-step linear gradient (4% to 13% B over 30 min, 13% to 20% B over 30 min, 20% to 28% B over 30 min, 28% to 35% B over 5 min, 35% to 95% B over 5 min) followed by a 5 min isocratic flush at 95% before washing and a return to low organic conditions.

Data acquisition was performed using a timsTOF pro2 system (Bruker Daltonics). 200 ng of each peptide sample were analyzed. The acquisition of data-dependent acquisition (DDA) runs was carried out with ion mobility in parallel accumulation serial fragmentation (PASEF) mode within a mobility range of 0.85–1.3. The ramp time was set at 100 ms with a duty cycle of 100%, and the scan range for mass (MS and MS/MS spectra) was set from 100 $m/z$ to 1700 $m/z$. The mobility-dependent collision energy ramping settings were optimized from 65 eV at an inverse reduced mobility (1/K0) of 1.3–20 Vs/cm² at 1/K0 of 0.6 Vs/cm², with a target intensity of 2000 and intensity threshold of 500.

The raw LFQ spectral files were searched against human human-reviewed Uniprot database using the LFQ-MBR workflow in Fragpipe 20.0 (MSFragger v4.0, IonQuant v1.10.12, Philosophor v5.1.0) to identify and quantify peptide abundance. In addition, trypsin and Lys-C were set as proteases, allowing up to two missed cleavages, methionine oxidation, and acetylation of N-termini were set as variable and carbamidomethylation of cysteine was set as a fixed modification. Both precursor and fragment mass tolerance were kept at 20 ppm with less than 1% FDR, and the

peptide length was kept between 7 and 50. Following the spectral analysis, the surface proteins were filtered using Uniprot-annotated membrane proteins. Proteins with at least one unique peptide were considered for further analysis.

FragPipe output data were analyzed using Metaboanalyst 5.0 or in R. Proteins annotated as "potential contaminant" or having missing values for 50% or more than 50% of the samples were filtered out, and missing values were imputed by KNN-feature-wise. Proteins were further filtered to include only surface-annotated proteins (same databases as for RNAseq analysis). Differentially expressed proteins were calculated using output from a Wilcoxon rank sum test after median normalization, comparing the log2-transformed relative protein abundance values between JMML and healthy donors with a false discovery rate set to 0.2 (Benjamini–Hochberg method).

Kernel density plots demonstrated similar levels of variability at the population level by comparing the distribution of bulk RNAseq expression levels and mass spectrometry protein abundance (Supplementary Fig. 3C). Limited sample-to-sample variability was observed in both bulk RNAseq and mass spectrometry datasets (Supplementary Fig. 3D).

### Generation of PDX

CAR T cell animal experiments were carried out in accordance with institutional guidelines approved by UCSF and complied with all relevant ethical regulations (application AN201693). NOD.Cg-Prkdc^scid Il2rg^tm1Wjl/SzJ (NSG) or NOD.Cg-*Prkdc^scid Il2rg^tm1Wjl* Tg (CMV-IL3, CSF2, KITLG)1Eav/MloySzJ (NSG-SGM3) were obtained from in-house breeding stocks at the UCSF Preclinical Therapeutics Core facility or from The Jackson Laboratory, housed and bred in a pathogen-free facility. A mixture of male and female mice was transplanted with 0.5-1e6 fresh or viably frozen patient PB or BM MNCs or unfractionated passaged PDX spleen cells via tail vein injection after two consecutive days of busulfan conditioning (25 mg/kg i.p.) and one day of rest. Tumor engraftment was assessed by flow cytometry for hCD45⁺ percentage in PB from submandibular bleeding. Upon termination, the patient's primary mutation was confirmed in PDX BM cells by Sanger sequencing.

### Murine experiments

Six to twelve-week-old mice were used unless otherwise stated. For PDX termination and survival studies, mice were allocated to different experimental arms upon successful engraftment, which was defined as reaching close to 0.5% or more human CD45⁺ cells in PB, following an accepted convention for xenograft[77,78]. Mice were distributed in a way that each arm had equal initial tumor burden. Mice allocated to different experimental groups were sex-matched unless otherwise mentioned. One day later, mice received CAR T cells (CD4/CD8 1:1 either mixed directly before injection or after thawing) via tail vein injection. Mice were sacrificed when showing clear clinical signs of distress and reaching humane endpoint. There were no instances in which this maximum was exceeded. Upon euthanization, cardiac blood, spleen, and BM were obtained for analysis.

NOD.Cg-Prkdcscid Il2rgtm1Sug Tg (SV40/HTLV-IL3, CSF2)10-7Jic/JicTac (NOG-EXL) were obtained from Taconic Biosciences.

For assessing engraftment potential of CD34⁺CD38⁻ CLL-1⁺ vs CLL-1⁻ cells, two passage 2 HM2130 mice were euthanized (one NOG-EXL, one NSG) and BM was sorted by flow-cytometry activated cell sorting (FACS) with the following antibodies (Supplementary Data 10–12): CLL-1 APC, CD34 AF 488, CD38 BV421 and hCD45 APC/Fire 750. Of the CD34⁺CD38⁻ cells, the NOG-EXL mouse had 0.9% CLL-1⁺ and 0% CLL-1⁻ cells, and the NSG mouse had 0% CLL-1⁺ and 0.1% CLL-1⁻ cells. Five hundred cells/mouse were injected into busulfan-conditioned (as above) 11-week-old female NOG-EXL mice.

For secondary transplantation, frozen BM cells from the HM2130 survival study were thawed with DNAseI. After incubation with an anti-CD3 antibody (OKT3) to prevent T cell engraftment and GVHD, 1e6 cells/mouse were injected into busulfan-conditioned 11-week-old female NOG-EXL mice.

For the U937 cell line model, male NSG mice were injected intravenously (IV) via the tail vein with 1.8e6 U937 cells stably expressing luciferase. Four days after tumor administration, tumor burden was quantified using bioluminescence, and animals were randomized into treatment groups with equal average tumor burden. On day five after tumor injection, mice were injected IV with 5e6 of CAR, empty CAR, or UTD T cells at a 1:1 ratio of CD4/CD8 T cells (mixed after thawing). Bioluminescence imaging was done weekly to assess tumor burden (Perkin Elmer In Vivo Imaging System, Caliper Life Sciences). Survival endpoint of the study was determined by signs of symptomatic illness in the animals and required veterinary protocols for humane euthanasia.

Sex considerations were beyond the scope of the study design. The maximal tumor burden permitted by the ethics committee was "tumor interfering with normal function," which was not exceeded during these studies. Housing conditions included 12 h of light and dark, temperature between 67° and 74° Fahrenheit, and humidity between 30% and 70%.

### Expression of CLL-1 on U937 cells after CLL1CART treatment in vivo

U937 cell line was procured from the DSMZ cell line biorepository (Braunschweig, Germany). Cells were regularly tested for mycoplasma contamination. Cells were cultured in RPMI cell culture medium containing 10% heat-inactivated FBS, 2 mM L-glutamine, and 100 U/mL penicillin/streptomycin.

Six to eight-week-old female NSG mice (Jackson Laboratories) were injected IV with 0.25e6 U937 cells. After 3 days, empty or CLL1CART (5e6/mouse, of which >30% were transduced) were administered IV. Mice were sacrificed 14 days after CART injection. Quantitative flow cytometric analysis of hCD45$^+$/CD3$^-$ cells in the spleen was performed. Surface level expression of CLL-1 per cell was calculated using PE surface quant beads and flow cytometric Quantibrite method (BD Biosciences, 340495).

### Statistical analysis

Unless specified in the text, samples were measured as biological replicates and Student's unpaired two-tailed t-test with assumption of Gaussian distribution was used to compare two means, a one-way ANOVA with Tukey's correction was used for comparison between more than two groups with one independent factor and two-way ANOVA followed by Tukey's post-hoc test was used to assess differences between groups with more than one factor. Log-rank Mantel–Cox test was used for survival curves. Data were presented as mean ± standard error of the mean (SEM) unless otherwise indicated. p-values ≤ 0.05 were considered to be statistically significant (*$p \leq 0.05$, **$p \leq 0.01$, ***$p \leq 0.001$, ****$p \leq 0.0001$). All statistical analyses outside RNAseq and mass spectrometry were performed using Prism version 10 software (GraphPad Software Inc.). Source data are provided as a Source Data file.

### Reporting summary

Further information on research design is available in the Nature Portfolio Reporting Summary linked to this article.

## Data availability

All JMML next-generation sequencing data generated and analyzed during the current study have been deposited in the database of Genotypes and Phenotypes (dbGaP) under accession phs002504.v3.p2 (UCSF Database for the Advancement of JMML) (https://www.ncbi.nlm. nih.gov/projects/gap/cgi-bin/study.cgi?study_id=phs002504.v3.p2). The RNAseq from healthy controls generated for this study can be obtained by contacting the corresponding author. The mass spectrometry proteomics data have been deposited to the ProteomeXchange Consortium via the PRIDE partner repository with the dataset identifier PDX052910 (https://www.ebi.ac.uk/pride/archive/projects/PXD052910). The five CB HSC bulk RNAseq samples used in this study are available in the GEO database under accession code GSE1118956 (https://www.ncbi.nlm.nih. gov/geo/query/acc.cgi?acc=GSE111895). The three external scRNAseq control samples used in this study are available in dbGaP under accession code phs002371.v6.p1 (https://www.ncbi.nlm.nih.gov/projects/gap/ cgi-bin/study.cgi?study_id=phs002371.v6.p1). The GTEx publicly available data used in this study are available on the GTEx website (file GTEx_Analysis_2017-06-05_v8_RNASeQCv1.1.9_gene_tpm on https://www. gtexportal.org/home/downloads/adult-gtex/bulk_tissue_expression). The remaining data are available within the Article, Supplementary Information, or Source Data that are provided with this paper.

## Code availability

Source code available from https://doi.org/10.5281/zenodo.14606594 (https://doi.org/10.5281/zenodo.14606594)[79].

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

## Acknowledgements

We would like to thank the families who participated in this study and their treating physicians. The CLL-1 sequence was kindly provided by Prof. Malcolm K. Brenner, Baylor College of Medicine, Houston, USA. We would like to acknowledge the contributions of the staff of the UCSF pediatric BM transplant unit for access to healthy donor cells (Dr. Christopher Dvorak, Ivette Gutierrez). We thank Dr. Corynn Kasap for helpful advice on CAR T cells; Dr. Sharon Chen for discussions on flow cytometry; Dr. Robbie Majzner and Dr. Malcolm Holterhus for advice on murine PDX studies and CD3 depletion; Dr. William Nyberg and Chris Chang for advice on TCR depletion; Dr. Brandom Holmes for assistance with Incucyte; Akul Naik, Regan Volk and Andrew Condon for assistance with mass spectrometry, and Gianina Wicaksono for advice on T cell isolation. This work was supported by the National Institutes of Health, National Cancer Institute grants 1U54CA196519 (J.R., M.L.L., B.B., and E.S.); R37 CA266550 (M.L.L., B.B., and E.S.), R50 CA274213 (A.G.L.), by the Mildred-Scheel post-doctoral scholarship from the German Cancer Aid (J.W.), Cookies for Kids' Cancer (E.S.), Leukemia & Lymphoma Society with support from the Mike & Sofia Segal Foundation (S.K.T. and E.S.), the Alex's Lemonade Stand Foundation (grant 23-28205), 'R' Accelerated Award (S.K.T. and E.S.), the Shaw Foundation (E.S.) and the California Cancer League (E.S.). J.W. received funding during participation in the Clinician Scientist Program TITUS—the First Thousand Days of Life—Frühe Prägung und Prävention by Else–Kröner–Fresenius–Stiftung and Hannover Medical School. S.K.T. is a Scholar of the Leukemia & Lymphoma Society and holds the Joshua Kahan Endowed Chair in Pediatric Leukemia Research at the Children's Hospital of Philadelphia. E.S. is the William Fries II Chair of Pediatric Oncology at Benioff Children's Hospital. Murine cell line xenograft studies were performed at the UCSF Helen Diller Family Comprehensive Cancer Center (HDFCCC) Preclinical Therapeutics Core facility. Flow cytometry and cell sorting were performed at the HDFCCC Laboratory for Cell Analysis Shared Resource Facility, supported through a grant from the NIH (P30CA082103).

## Author contributions

J.W. and E.S. designed and conceived the study and prepared the manuscript. J.W. designed the approach of and performed experiments, and analyzed/interpreted data. C.Z. and S.M. performed RNAseq. J.W., E.R., and A.B. performed and analyzed mass spectrometry data. A.G.L., C.Z., and J.M. performed bioinformatics analysis. S.X., S.A., and C.M. processed patient samples. S. Bachl performed Incucyte on A375 and assisted with *TRAC*-KO. J.W., S.A. S.X., J.R., K.Y., H.S., S. Bhatnagar, and J.K.D. performed murine and flow cytometric experiments. S.A., H.S., and B.P.E. performed FACS. B.P.E., K.M., A.I., S.C., W.C.T., M.L.L., and B.B. contributed to experimental design and provided critical experimental resources. S.K.T. and A.P.W. interpreted experimental data and edited the manuscript. A.P.W. and E.S. supervised the study. All authors reviewed and approved the final manuscript.

## Competing interests

The authors declare no competing interests.
