## [Transparent Peer Review file · Nature Communications]

Cellular immunotherapy targeting CLL-1 for juvenile myelomonocytic leukemia

Corresponding Author: Dr Elliot Stieglitz

Version 0:

Reviewer comments:

Reviewer #1

(Remarks to the Author)

To the authors,

Werner, et al present an interesting study in which they systematically analyzed bulk RNA and single cell RNA sequencing datasets for associated surface proteins that could be candidates for chimeric antigen receptor (CAR) T-cell therapy against juvenile myelomonocytic leukemia (JMML). Through the robust screening, out of 15 targets that were more highly expressed in JMML compared to healthy controls, they ultimately chose to study CLEC-12A (CLL-1) because it was the furthest in clinical development, and validated its protein expression. They constructed 4 different CLL-1 CAR-eGFP fusion proteins, testing different combinations of hinge and endodomains, and confirmed that the CD8 hinge + transmembrane domain and the 41BB-zeta endodomain combination was the most effective.

They subsequently demonstrated that T-cells transduced with this CLL-1 CAR construct showed efficacy against JMML cell lines and primary patient cells in vitro and in vivo, and furthermore showed that efficacy was preserved in TRAC knockout T-cells.

This work validates prior studies with CLL-1 CAR T-cells against myeloid leukemia. Several minor revisions/comments are raised.

1. For Figure 6: can you please provide flow cytometric assessment of the JMML burden over time, instead of only at the Day 59 timepoint, and provide the Kaplan Meier curve for this study? It would be a useful comparison to the information showed in Figure 7 for the HM2130 PDX model, and to the U937 histiocytic lymphoma model.
2. Can you comment on whether there is any decrease in mean fluorescence intensity (MFI) of CLL-1 expression in the analyzed remaining U937 cells and primary cells in mice at the end of the experiments in Figures 5, 6, and 7, compared to CLL-1 expression of the cells in culture or during earlier timepoints of the experiment? It would be interesting to see if CLL-1 is subject to antigen downregulation/antigen escape in JMML cells.
3. For Figure 3, since normal mature myeloid cells including monocytes and dendric cells should express CLL-1, why wasn't any CLL-1 expression seen in bone marrow or peripheral blood of healthy donors? Can you provide the gating strategy?
4. Please include a graph demonstrating the transduction efficiencies for your CLL-1 CAR eGFP fusion constructs.

(Remarks on code availability)

Reviewer #2

(Remarks to the Author)

This paper aims to provide key pre-clinical data for the utility of a CART against CLL1 for the treatment of children with JMML, either as a bridge to transplantation or in the relapse setting. The overall aim of this work aims to address a major unmet clinical need for this disease. Using a multiomic approach the authors identify CLL1 as a differentially expressed cell surface target in JMML at transcript and protein level. CARTs are developed and validated in experiments carried out in cell line and PDX models.

MAJOR COMMENTS

CLL1 is already a well-recognised cell surface target in myeloid malignancies with CART cells already in clinical development. It is not really a major step forward to show this target is also overexpressed in JMML. The potential novelty lies in whether the authors can convincingly demonstrate differential expression and targetability of CLL1 in stem cell populations. Here the study falls short.

The differential expression of CLL1 on the cell surface of bona fide stem cells is unclear. For example, Fig 3D seems to show no differential expression of CLL1 by FACS in JMML vs control. There is also a very low level of positivity for CLL1 by in CD34+CD38- cells where most LSC activity resides in JMML e.g. extended Fig 3C shows most JMML have <5% positivity by FACS in this compartment. I suppose it is possible (although unlikely) that these rare cells are bona fide LSCs, which would need to be demonstrated by purifying CLL1+ vs CLL1- cells from the CD34+CD38- compartment and assessing LSC activity in each compartment in a xenograft assay

SPECIFIC COMMENTS

Figure 2 and Extended Figure 2

Authors should provide the gene list driving the clusters and specifically justify that clusters 10,11,20 represent HSC (more likely HSPC). The differential expression of CLL1 in putative HSPC should be shown as a violin plot (and across all clusters in control vs JMML). On the overlay CLL1 appears to only be expressed on a subset of cells (see also below). The MS data also appears very noisy and suggests a high degree of pt-to-pt variability. This should be shown in supplemental i.e. how CLL expression varies across pts in MS and transcript data.

Figure 3

Data presented in figure 3 and extended data figure 3 do not currently support clear differential expression of CLL1 in HSPC compartment. Authors should show the full gating strategy and CLL1 expression by both flow plots and describe fully in the methods. Why do the authors think CLL1 is a good target when it is only expressed on a minority of CD34+ cells?

Figure 7

How is it possible that there is such a major reduction in JMML cells by FACS, including CD34 compartment, when only a minority of cells express the target? With such a major reduction in disease burden, why is survival of mice only marginally improved e.g. Fig 7H vs I. Secondary transplantation is required to demonstrate reduction/eradication of LSC (which seems unlikely if mice are dying of leukaemia). Healthy control xenografts are required as an additional control to demonstrate specific targeting of JMML and not healthy stem cells.

(Remarks on code availability)

Reviewer #3

(Remarks to the Author)

In the manuscript "Discovery and development of CLL-1 as a cellular immunotherapy target in JMML", the teams of Elliot Stieglitz and Arun P. Wiita identified CLL-1 as a targetable surface marker on JMML cells. They also developed several CLL-1 directed CAR-T cells and showed their efficacy against JMML, both in vitro and in a PDX mouse model. The use of immunotherapy in JMML is very interesting, and the combined use of bulk and single cell RNA sequencing, mass spectrometry and flow cytometry represents a powerful approach. Due to the poor outcome of high risk JMML patients, there is a high medical need for novel therapies. However, there are several limitations of this study precluding publication in the current form. The major weakness of the manuscript is the over/misinterpretation of some results, especially the statement that leukemia stem cells (LSC) are successfully hit by CLL1CART. Indeed, none of the experiments performed in this manuscript addressed LSCs or HSCs.

Major points:

- Throughout the manuscript, LSCs were defined as CD34+38+ cells. However, it has been shown earlier that JMML-LSCs are CD34+38- (Louka et al, JEM, 2021). Results shown in Figure 3D (direct costaining with CD33) and in Figure 4G (indirect evidence by use of CD33CART) indicate that the CD34+ cells analyzed by the authors rather are differentiating, myeloid progenitors. The authors should show the expression of CLL-1 on CD34+38- cells (ideally costained with CD90), demonstrate depletion of these cells by CLL1CART treatment in vitro and in PDX. In addition, serial transplantation is required to confirm that JMML-LSCs treated with CLL1CART were successfully depleted. Only such experiments can determine the effects of CLL1CART on LSCs.

- Introduction: CLL-1 should be introduced including its function, physiological expression and earlier use of CLL1CART.

- Results: CLL-1 expression on JMML MNCS: the CLL-1 expression is very variable. The authors should include some information on the expression of CLL-1 in the different methylation groups (LM-IM-HM). Since novel therapies are only required for IM/HM JMML patients, this information is crucial.

- Figure 3A: How was the CLL1-MFI determined? To determine upregulation of an individual cell, the MFI should be determined only in CLL-1 positive cells. If the whole cell population was analyzed, the higher MFI levels simply reflects the higher amount of CLL-1 positive cells. In addition, Figure 3A would benefit from a logarithmic presentation.

- Data shown in Figure 4 are only technical replicates. At least 3-4 patient samples should be used to demonstrate that the effect is reproducible. For Figure 4C-D, raw data instead of normalized values should be shown.

- It is not fully clear, when the PDX mice were treated. Figure 6A and 7A indicate that treatment was initiated 4 weeks after xenotransplantation. This is a very early time point to treat JMML in a PDX, given that JMML results in a slow, chronic disease in xenotransplanted mice. The authors should discuss this.

- Discussion line 263: please remove the word "specifically". As stated by the authors, CLL1CART were used already for other disease entities (lines 308 etc.).

- Line 287: The authors should omit the statement that CLL1CART is superior over CD33CART. Their in vitro data indicate that CD33CART is more potent than CLL1CART (Figure 4), and no direct comparison was performed in vivo.

Minor points:

- Why are JMML cells defined as "GFP-negative"?

- Sentence in line 235-236 is not correct (First, an NRAS-mutant JMML PDX model...).

(Remarks on code availability)

Version 1:

Reviewer comments:

Reviewer #1

(Remarks to the Author)

To authors,

The revised version has clearer message and the authors have addressed my concerns decidedly. I have no more requests to ask.

Reviewer #2

(Remarks to the Author)

I appreciate the efforts made by the authors to address my comments.

The new secondary transplant data is interesting and a nice addition. However, it is unclear if any of the mice developed leukemia in secondary transplant. What did the CLL-KO mice die of in survival analysis?

The flow cytometry data essentially confirm my previous concern that CLL1 is not overexpressed on the surface of JMML HSPC, nor are CLL1 positive cells increased as a proportion of CD34 compartment. I do not think it is correct to state that CLL1 is "frequently overexpressed on LSC" (line 311), the data do not support this. Absolute numbers of CLL1 expressing cells are increased versus control (particularly in PB) due to the expansion of myeloid cells and precursors as part of the disease phenotype. This is why the RNA-seq and MS data identify CLL1 overexpression. Further, CLL1 negative cells from JMML pts engraft mice. Collectively, this does not make CLL1 a particularly compelling target in my view and yet the functional data is convincing.

I read the arguments made by the authors relating to the disconnect between expression of CLL1 and efficacy of CLL1 targeting CARTs. The comparison with CD19 is for me not convincing. For example, it is very clear that CD19 -ve clones are selected for with CD19 directed CART treatments confirming a very clear link to expression of the target.

I do not wish to hold up publication of this work and would suggest that the authors clearly acknowledge that the expression of CLL1 on HSPC is not different to control HSPC, and that CLL1 is only expressed only on a minority of JMML cells in CD34 compartment (not different to control) and only about 50% of MNC (with a lot of inter-patient heterogeneity, so the reason why CART targeting of CLL1 so effectively eliminates all of JMML cell engraftment is uncertain and requires further study; the authors could speculate on the possible explanation in the discussion.

Additional point:

The data in extended Figure 3C is very oddly distributed for MFI data. It looks bimodal with over 1 log difference between the groups, without explanation.

Reviewer #3

(Remarks to the Author)

Werner et al greatly improved their manuscript and met all my concerns.

We appreciate the constructive feedback of all three reviewers. In brief, we agree with all the reviewer's suggestions and thank them for improving the clarity and impact of our paper. Our detailed responses are below:

REVIEWER COMMENTS

Reviewer #1

To the authors,

Werner, et al present an interesting study in which they systematically analyzed bulk RNA and single cell RNA sequencing datasets for associated surface proteins that could be candidates for chimeric antigen receptor (CAR) T-cell therapy against juvenile myelomonocytic leukemia (JMML). Through the robust screening, out of 15 targets that were more highly expressed in JMML compared to healthy controls, they ultimately chose to study CLEC-12A (CLL-1) because it was the furthest in clinical development, and validated its protein expression. They constructed 4 different CLL-1 CAR-eGFP fusion proteins, testing different combinations of hinge and endodomains, and confirmed that the CD8 hinge + transmembrane domain and the 41BB-zeta endodomain combination was the most effective.

They subsequently demonstrated that T-cells transduced with this CLL-1 CAR construct showed efficacy against JMML cell lines and primary patient cells in vitro and in vivo, and furthermore showed that efficacy was preserved in TRAC knockout T-cells.

This work validates prior studies with CLL-1 CAR T-cells against myeloid leukemia. Several minor revisions/comments are raised.

1. For Figure 6: can you please provide flow cytometric assessment of the JMML burden over time, instead of only at the Day 59 timepoint, and provide the Kaplan Meier curve for this study? It would be a useful comparison to the information showed in Figure 7 for the HM2130 PDX model, and to the U937 histiocytic lymphoma model.

We thank the reviewer for this suggestion. We have now added flow cytometric assessment data from day 33 and day 46 to the existing data point from day 61 to Fig. 6B:

We realize that we were not sufficiently clear in the text that we intentionally planned from the start of this experiment to sacrifice all the mice at the same timepoint (what we refer to as a termination study) as opposed to sacrificing the mice when they each reach endpoint at different times (what we refer to as a survival study). There are advantages and disadvantages to each type of experiment. Figure 6 was a termination study focusing on percent engraftment of human cells in cardiac blood, bone marrow and spleen tissue. A Kaplan Meier survival estimate for this termination study is displayed below. We have now clarified this in the text (lines 264-267 of the manuscript).

Response letter Fig. 1 – not included in manuscript

2. Can you comment on whether there is any decrease in mean fluorescence intensity (MFI) of CLL-1 expression in the analyzed remaining U937 cells and primary cells in mice at the end of the experiments in Figures 5, 6, and 7, compared to CLL-1 expression of the cells in culture or during earlier timepoints of the experiment? It would be interesting to see if CLL-1 is subject to antigen downregulation/antigen escape in JMML cells.

The reviewer raises an important question about antigen escape, a potential mechanism of resistance to CART cell therapy. To address this concern, we have performed the following analyses:

- (1) We have now analyzed the MFI for CLL-1 in engrafted CLL-1⁺ human cells (gated live hCD45⁺ CD3/8⁻ CLL-1⁺) of the HM5896 termination study (Supplementary Data Fig. 5C and lines 285-287 of the manuscript). We did not observe a significant reduction in the MFI of CLL-1 in the CLL-1CART treated mice for endpoint tissues.

- (2) We also performed the same analysis for the HM2130 termination study (Supplementary Data Fig. 5D and lines 285-287 of the manuscript) and observed nominal changes across various tissues.

(3) For the HM2130 survival study we also performed quantification of CLL-1 on the surface of CLL-1⁺ cells in bone marrow upon mice reaching humane endpoint (Supplementary Data Fig. 5B and lines 285-287 of the manuscript). CLL-1 quantification did not indicate CLL-1 antigen downregulation in CLL1CART treated mice versus control.

(4) In regards to U937 experiments, we had not cryopreserved tissues from our original experiments (Fig. 5) so we were unable to assess for changes in MFI. We therefore performed a new experiment in which we treated a U937 CDX with CLL1CART and did not observe significant changes in the MFI of CLL-1 of hCD45⁺CD3⁻ U937 cells in the spleen two weeks after T cell injection when the study was terminated (Supplementary Data Fig. 3 and lines 243-245/676-686 of the manuscript).

Supplementary Data Fig. 3

Overall we did not observe significant changes in the MFI of CLL-1 in our PDX or CDX experiments to suggest antigen escape.

3. For Figure 3, since normal mature myeloid cells including monocytes and dendritic cells should express CLL-1, why wasn't any CLL-1 expression seen in bone marrow or peripheral blood of healthy donors? Can you provide the gating strategy?

We agree with the reviewer that normal mature myeloid cells including monocytes, dendritic cells and granulocytes typically express CLL-1. Figures 3A and 3C display control bone marrow and peripheral blood cells and the expression of CLL-1 is present in the cells mentioned. Interestingly, the MFI ratio for CLL-1 is higher in bone marrow compared to peripheral blood. To make this clearer for the readers, the gating strategy is now provided in Supplementary Data Fig. 8 ((A) depicts FMOs, (B) depicts the primary sample, both HM4905):

Supplementary Data Fig. 8

A FMOs with HM4905:

B HM4905

4. Please include a graph demonstrating the transduction efficiencies for your CLL-1 CAR eGFP fusion constructs.

The transduction efficiencies of the different CLL-1 CAR eGFP fusion constructs are now shown in Supplementary Data Fig. 17B-C (lines 522-523 of the manuscript):

There is a trend towards increased transduction efficiency with higher multiplicity of infection (MOI) for CD8 T cells compared to CD4 T cells which are generally easier to transduce.

Reviewer #2

This paper aims to provide key pre-clinical data for the utility of a CART against CLL1 for the treatment of children with JMML, either as a bridge to transplantation or in the relapse setting. The overall aim of this work aims to address a major unmet clinical need for this disease. Using a multiomic approach the authors identify CLL1 as a differentially expressed cell surface target in JMML at transcript and protein level. CARTs are developed and validation experiments carried out in cell line and PDX models.

MAJOR COMMENTS

CLL1 is already a well-recognised cell surface target in myeloid malignancies with CART cells already in clinical development. It is not really a major step forward to show this target is also overexpressed in JMML. The potential novelty lies in whether the authors can convincingly demonstrate differential expression and targetability of CLL1 in stem cell populations. Here the study falls short.

We agree with the reviewer that demonstrating activity of CLL-1 CAR T cells in stem cell populations of JMML would be of great interest to the community and per the reviewer's suggestions below (Fig. 8 and page 22 of this document), we have performed additional experiments that we believe address these concerns and strengthen the manuscript. That being said, there are currently no clinical trials for CLL-1 CAR T cells that allow for the inclusion of

JMML patients (or any CAR T study for that matter). We believe that this manuscript as a whole will lead to the development of trials for this orphan disease.

The differential expression of CLL1 on the cell surface of bona fide stem cells is unclear. For example, Fig 3D seems to show no differential expression of CLL1 by FACS in JMML vs control. There is also a very low level of positivity for CLL1 by in CD34⁺CD38⁻ cells where most LSC activity resides in JMML e.g. extended Fig 3C shows most JMML have <5% positivity by FACS in this compartment. I suppose it is possible (although unlikely) that these rare cells are bona fide LSCs, which would need to be demonstrated by purifying CLL1⁺ vs CLL1⁻ cells from the CD34⁺CD38⁻ compartment and assessing LSC activity in each compartment in a xenograft assay.

We were not sufficiently clear about the content of Extended Data Fig. 3C and have now clarified in the text (lines 187-191 of the manuscript) that the figure demonstrates the percentage of CD34⁺CD38⁻ or CD34⁺CD38⁺ cells within JMML or control samples, not the CLL-1 expression on CD34⁺CD38^{-/+} cells.

Per the reviewer's excellent recommendation, we have now performed the data analysis that the reviewer suggested for bone marrow (Extended Data Fig. 3E).

We observed heterogenous expression of CLL-1 on CD34⁺CD38^{-/+} cells but observed trends towards higher expression in JMML CD34⁺CD38⁺ bone marrow cells (that also have LSC properties as per the Louka et al. paper, lines 82-85 of the manuscript) versus controls. CLL-1 expression on CD34⁺CD38⁻ cells was more varied compared to healthy controls. We tempered our comments throughout the manuscript (lines 88-90/134/165-166/191-193 of the manuscript) and acknowledge that CLL-1 is not uniformly overexpressed on CD34⁺CD38⁻ LSCs.

As expected, CD34⁺ cells from control peripheral blood did not yield an adequate number of cells for subsetting into CD34⁺CD38⁻ and CD34⁺CD38⁺ cells (Extended Data Fig. 3F). Nevertheless, JMML CD34⁺ cells from peripheral blood have a higher mean percentage of CLL-1⁺ cells compared to controls (Fig. 3D).

We followed the reviewer's suggestion and sorted CLL-1⁺ versus CLL-1⁻ CD34⁺CD38⁻ cells from mice as outlined in the additional "Methods" paragraph (lines 657-663 of the manuscript): "For assessing engraftment potential of CD34⁺CD38⁻ CLL-1⁺ versus CLL-1⁻ cells two passage 2 HM2130 mice were euthanized (one NOG-EXL, one NSG) and bone marrow was sorted by flow-cytometry activated cell sorting (FACS) with the following antibodies (Supplementary Data Tables): CLL-1 APC, CD34 AF 488, CD38 BV421 and hCD45 APC/Fire 750. Five-hundred cells/mouse were injected into three busulfan-conditioned (as above) 11-week-old female NOG-EXL mice/group."

The gating strategy for sorting is outlined in Supplementary Data Fig. 18:

Supplementary Data Fig. 18

Mice in both groups engrafted with a trend towards higher engraftment for the CLL-1⁺ cells as measured in peripheral blood (Supplementary Data Fig. 6A-C and lines 290-294 of the manuscript). Upon termination 83 days post LSC injection, the CLL-1⁺ group showed a trend towards higher engraftment (Supplementary Data Fig. 6D). Based on these experiments, CLL-1 is not required for engraftment of LSCs. This is similar to pediatric AML where not all CD34⁺CD38⁻ cells express CD33⁺, hence CD33 is dispensable for tumor maintenance however still presents a potentially suitable therapeutic target in myeloid leukemias with high expression on blasts and LSCs.

Supplementary Data Fig. 6

Most importantly, as we now show in the manuscript (Fig. 8 and page 22 of this document), targeting CLL-1 with CAR T cells in mice does empirically deplete LSCs resulting in decreased serial transplantability (lines 59-60/103-105/316-320 of the manuscript).

SPECIFIC COMMENTS

Figure 2 and Extended Figure 2

Authors should provide the gene list driving the clusters and specifically justify that clusters 10,11,20 represent HSC (more likely HSPC). The differential expression of CLL1 in putative HSPC should be shown as a violin plot (and across all clusters in control vs JMML). On the overlay CLL1 appears to only be expressed on a subset of cells (see also below).

We appreciate the reviewer's concerns regarding the need for more convincing evidence to demonstrate that clusters 10, 11, and 20 (which we have collectively labeled as "HSC" in our manuscript and now re-labeled "HSPC") in fact contain hematopoietic stem and progenitor cells (HSPCs). As outlined in the Methods section, we initially utilized ScType to automate the assignment of HSCs among other cell types. The specific gene markers used for identification have now been added to Supplementary Data Table 14, along with markers for other cell types as suggested by the reviewer. While ScType provided a frequently cited and automated approach, we also manually verified these clusters by visualizing both the percentage of cells expressing key canonical markers and their expression levels. These markers include CD34, CD38, CD117 (KIT), and CD133 (PROM1)². This data was originally presented in Extended Data Fig. 2A, and we have now expanded it to include GATA2, a critical gene for HSC self-renewal and maintenance³. While no marker in and of itself defines a HSC, by visualizing these main canonical markers together, we can confirm their prominent expression in clusters 10, 11, and 20 with respect to the other clusters, thereby reinforcing their identification.

As suggested by the reviewer, to further refine our characterization of these clusters, we have now referenced the immunophenotypic markers for HSPCs as described by Louka et al.⁴. While Louka et al. used flow cytometry for cell sorting, we adapted these marker combinations to assess the HSC and HSPC populations within our scRNAseq data, as detailed in the table below (Extended Data Fig. 2A and lines 419-424 of the manuscript). Here we employed the scGate algorithm⁵. This is an automated signature scoring method allowing us to construct a synthetic gating model for each hematopoietic lineage for individual cells, enabling a more granular analysis cell-by-cell instead of labeling the entire clusters, e.g., 10, 11, and 20 as with ScType. Extended Data Fig. 2B below shows a stacked plot to visualize the tabulation of each cell type within its respective clusters. From this analysis, we were able to detect 3 additional HSPC: GMPs, LMPPs, and MPPs.

Extended Data Fig. 2

A

Cell.Type	Description	Marker.panel
HSCs	(Hematopoietic Stem Cells)	Lin-CD34+CD38-CD90+CD45RA-
HSC-abberant	(Abberant Hematopoietic Stem Cells, Louka 2021)	Lin-CD34+CD38-CD90+CD45RA+
MPPs	(Multipotent Progenitors)	Lin-CD34+CD38-CD90-CD45RA-
LMPPs	(Lymphoid Primed Multipotent Progenitors)	Lin-CD34+CD38-CD90-CD45RA+
CMPs	(Common Myeloid Progenitors)	Lin-CD34+CD38+CD123+CD45RA-
MEPs	(Megakaryocyte Erythroid Progenitors)	Lin-CD34+CD38+CD123-CD45RA-
GMPs	(Granulocyte Monocyte Progenitors)	Lin-CD34+CD38+CD123+CD45RA+

B

Moreover, as per the reviewer's suggestion, we plotted the expression of *CLEC12A* (CLL-1) across the HSPC cluster (containing clusters 10, 11 and 20) as well as clusters 10, 11, and 20 individually using violin plots (Extended Data Fig. 2C). This visualization not only enables a clear representation of *CLEC12A* expression within these putative HSC/HSPC clusters but also allows for a direct comparison between control and JMML samples. We also created a similar plot from the HSCs and their progenitor populations (Extended Data Fig. 2D and lines 137-141 of the manuscript), further illustrating the differential expression of *CLEC12A* across various HSC and HSPC subtypes in both normal and JMML conditions. Collectively, these additional analyses demonstrate that clusters 10, 11, and 20 comprise HSCs with potential heterogeneity, including HSC-derived HSPCs. Notably, analysis of *CLEC12A* reveals significant upregulation across all these cell types, supporting *CLEC12A*'s potential as a therapeutic target in JMML, irrespective of the specific HSC or their progenitor cell type.

The MS data also appears very noisy and suggests a high degree of pt-to-pt variability. This should be shown in supplemental i.e. how CLL expression varies across pts in MS and transcript data.

We agree with the reviewer and appreciate the opportunity to expand on interpatient variability and now show CLL-1 expression across patients in mass spectrometry data (Extended Data Fig. 2H, lines 164-165 of the manuscript).

Regarding the variability in our MS data and the need to compare this with the transcript data, we conducted additional analyses to quantify and visualize the variability in both MS and RNAseq datasets. **First**, we compared the distribution of RNAseq expression levels and MS protein abundance using kernel density estimation (Supplementary Data Fig. 1C and lines 631-634 of the manuscript). While the scales for MS and RNAseq data are

not directly comparable the density plot reveals that both MS and RNAseq data exhibit similar overall distribution shapes, with RNAseq showing a slightly wider range of values, suggesting that both techniques capture similar levels of variability at the population level. **Secondly**, we conducted a correlation analysis between the average values of RNAseq and MS for each gene across all samples within each datatype separately. These values were then presented as a scatter plot, with RNAseq data on the x-axis and MS data on the y-axis (Supplementary Data Fig. 1A and lines 171-172 of the manuscript). Each point on this scatter plot represents a gene or protein, with its position indicating its average values in both datatypes. We fitted a linear regression line to assess the overall trend and calculated the Pearson correlation coefficient. The statistically significant positive correlation (Pearson's $r = 0.42$, $p < 0.001$) between MS and RNAseq measurements demonstrates the consistency of our findings across proteomics and transcriptomics platforms. The relationship between mRNA and protein levels is consistent with other datasets that have compared proteome versus transcriptome abundance ($R^2=0.41$ between mRNA and protein abundances)⁶. **Third**, to illustrate the variability across all patient samples, we created violin plots showing the distribution of protein abundance or expression levels for each sample in both MS and RNAseq datasets (Supplementary Data Fig. 1D and lines 631-634 of the manuscript). This plot reveals the extent of variability within and between samples for both data types. Again, it is important to note that the scales for MS and RNAseq data are different and not directly comparable. However, **this visualization highlights the limited sample-to-sample variability in both datasets. We hope this addresses the reviewer's question regarding the degree of variability** in the MS data while also providing context by comparing it to the RNAseq data distribution patterns. **Fourth**, while we only had three matched patient samples with both MS and RNAseq data available, we performed correlation analyses on these samples to further investigate the relationship between protein abundance and mRNA expression (Supplementary Data Fig. 1B and lines 171-172 of the manuscript). Each panel represents a different patient (HM4326, HM4905, and HM5255), showing the correlation between MS levels and RNAseq expression for individual genes. All three patients show statistically significant positive correlations ($p < 0.001$ for all), with Pearson's r values ranging from 0.35 to 0.46. Despite the limited number of matched samples, this analysis speaks to the overall consistency between MS and mRNA expression at the individual patient level, while also revealing some patient-to-patient variability and their relationship. Together, these analyses demonstrate that while the MS data exhibits some patient-to-patient variability, its overall distribution patterns are comparable to those of RNAseq data (Supplementary Data Fig. 1C-D). Importantly, the significant positive correlations between MS and RNAseq measurements, both across all samples (Supplementary Data Fig. 1A) and in matched patient samples (Supplementary Data Fig. 1B), indicate that despite the inherent variability, there is a consistent relationship between protein abundance and mRNA expression levels. Thus, the observed variability in MS data, rather than being a limitation, likely reflects the biological complexity and heterogeneity of *CLEC12A* (CLL-1) across patients.

Supplementary Data Fig. 1

Figure 3

Data presented in figure 3 and extended data figure 3 do not currently support clear differential expression of CLL1 in HSPC compartment. Authors should show the full gating strategy and CLL1 expression by both flow plots and describe fully in the methods.

We agree with the reviewer and have now added the full gating strategy and describe our methods. A description of the gating strategy of CLL-1 expression is now found in the Methods (lines 465-470 of the manuscript): “Flow cytometric analysis for CLL-1 expression included the following gates in subsequent order: cells of interest in FSC-A/SSC-A gate, doublet exclusion in FSC-A/FSC-H gate, dead cell exclusion in live dead (LD) plot, selection of hCD45⁺ cells and then either CLL-1 versus CD33 expression or CD34 versus CD38 expression. From the latter, CLL-1 versus CD33 expression was also determined in different subpopulations of CD34^{+/-} and CD38^{+/-} cells.”

A representative gating strategy including primary JMML samples can be found in Supplementary Data Fig. 8 (on page 5 of this document) and Supplementary Data Fig. 9-12 (see below) and Supplementary Data Fig. 13 (on page 20 of this document).

Supplementary Data Fig. 9

HM2616

Supplementary Data Fig. 10

HM2875

Supplementary Data Fig. 11

HM5886

Supplementary Data Fig. 12

HM4601

Why do the authors think CLL1 is a good target when it is only expressed on a minority of CD34+ cells?

We thank the reviewer for the opportunity to address this important question.

In summary, we decided to pursue CLL-1 as CAR T cell target for JMML due to multiple reasons (lines 310-315 of the manuscript):

1. overexpression in the JMML bulk mononuclear cell population by RNAseq and flow cytometry versus healthy controls which would allow for reduction of the bulk tumor population
2. overexpression in JMML HSPC populations by scRNAseq versus healthy controls which includes cells of the LSC compartment (HSCs, GMPs) (Extended Data Fig. 2C-D and page 11 of this document)
3. overexpression in JMML HSPCs (BM CD34⁺ and CD34⁺CD38⁺ cells as well as PB CD34⁺ cells by flow cytometry versus healthy controls which includes cells of the LSC compartment (no further subsetting in PB was possible due to low stem cell content in healthy controls) (Fig. 3 and Extended Data Fig. 3E-F (page 7 of this document))
4. CLL-1 was the only target that was overexpressed in HSPCs in both scRNAseq and mass spectrometry of CD34⁺ cells versus healthy controls (Fig. 2C) that also fulfilled criteria 5 and 6 (see below)

5. low expression of the chosen target in healthy tissue as per GTEx database
6. CLL-1 CAR T cells are already being clinically investigated in adult and pediatric AML patients which might make transfer to the orphan disease JMML with very few patients more feasible

We acknowledge that flow cytometry demonstrates less differential expression on CD34⁺CD38⁻ cells between JMML and controls. However high expression on HSPCs is not the only consideration for target choice as other factors such as on-target off-tumor toxicity are also relevant. For example, CD33 CAR T cell therapy has been associated with life threatening infections due to its high expression on HSPCs.

We were able to demonstrate efficacy of CLL1CART against JMML HSPCs *in vitro* and *in vivo* for multiple patients. In addition, we now share additional data below that demonstrates that CLL1CART empirically target the LSC population as evidenced by dramatically lower secondary transplantability *in vivo* of CLL1CART treated cells compared to appropriate controls (Fig. 8 and on page 22 of this document).

Lastly, there is most likely interpatient heterogeneity for CLL-1 expression and the decision to treat an individual patient may need to be based on a minimum threshold of CLL-1 expression. This requirement is currently in place in an AML clinical trial (that excludes patients with JMML) where the inclusion criteria require >30% positivity of CLL-1 on blasts by flow cytometry or immunohistochemistry (NCT04219163). We now mention this in the manuscript text on lines 338-340.

Figure 7

How is it possible that there is such a major reduction in JMML cells by FACS, including CD34 compartment, when only a minority of cells express the target? With such a major reduction in disease burden, why is survival of mice only marginally improved e.g. Fig 7H vs I.

This is an important question although we would respectfully acknowledge that there are many examples of immunotherapies being remarkably effective even when the target is not expressed uniformly. This applies to FDA-approved CD19 CAR T cells⁷ (“The response rates did not appear to be influenced by biologic covariates, such as the prevalence and intensity of CD19 expression”), FDA approved antibody drug conjugates⁸ (“Remission rates were significantly higher with inotuzumab ozogamicin than with standard therapy among patients with both higher ($\geq 90\%$) and lower ($< 90\%$) levels of CD22 expression”) and others⁹ (“Baseline BCMA intensity by flow cytometry did not correlate with either expansion or response in our study, suggesting that excluding patients based on baseline BCMA expression is likely not necessary.”). There are several possible explanations for this phenomenon including bystander effect, killing of antigen negative cells because they are sensitive to CAR T cell-derived cytokine-induced apoptosis¹⁰, and the possibility that the target is actually present but below the level of quantification by whatever technique is being employed, but we believe this is outside of the scope of our manuscript. More directly related to our manuscript, Fig. 7 which uses HM2130 has 68.8% of bulk MNCs, 47.4% of CD34⁺ cells, 27.2% of CD34⁺CD38⁻ cells and 68.6% of CD34⁺CD38⁺ cells express CLL-1, respectively (see above graph “FMOs with HM4905” in Supplementary Data Fig. 8 and on page 5 for FMOs, see Supplementary Data Fig. 13 for HM2130 primary patient PB sample). This is substantial expression of CLL-1 which explains the major reduction in JMML cells measured by flow cytometry. Most importantly, we thank the reviewer for encouraging us to perform secondary transplantation experiments.

Supplementary Data Fig. 13

The reviewer raises an important question regarding the cause of death in the survival study using HM2130 which we investigated further by thawing bone marrow from mice that reached humane endpoint in this study and assessed it for tumor burden by flow cytometry. We observed that CLL1CART treated mice had evidence of leukemia as well as CD34⁺ cells, therefore they likely died of JMML. However burden of these two cell populations showed a trend towards lower burden in CLL1CART versus UTD and PBS (Supplementary Data Fig 5). CLL-1 quantification did not indicate CLL-1 antigen downregulation in CLL1CART treated mice versus control (Supplementary Data Fig. 5B and lines 284-285 of the manuscript).

The HM2130 survival study used two subsequent doses of 1 and 10e6 CLL1CART whereas the HM2130 termination study used one dose of 5e6, therefore they are not directly comparable. Given the data, we hypothesize that choosing a high enough dose earlier in the disease course will lead to better response.

Supplementary Data Fig. 5

Secondary transplantation is required to demonstrate reduction/eradication of LSC (which seems unlikely if mice are dying of leukaemia).

We followed this excellent suggestion which we now describe in the additional “Methods” paragraph (lines 664-667 of the manuscript): “For secondary transplantation frozen bone marrow cells from the HM2130 survival study were thawed with DNaseI. After incubation with an anti-CD3 antibody (OKT3) to prevent T cell engraftment and GVHD, 1e6 cells/mouse were injected into busulfan-conditioned (as above) 11-week-old female NOG-EXL mice.”

CLL1CART mice had lower tumor burden in peripheral blood over time (Fig. 8A-B, Extended Data Fig. 10A) as well as significantly lower spleen length and weight (Fig. 8C). This corresponded to significantly lower JMML burden in bone marrow, spleen and cardiac blood (Fig. 8D) upon termination 61 days after bone marrow injection as well as lower HSPC burden (Fig. 8E-G, Extended Data Fig. 10B-D) (lines 289/295-302 of the manuscript).

Despite likely heterogeneity in CLL-1 expression in LSC containing compartments, we observed a dramatic reduction in serial transplantability of CLL1CART treated cells *in vivo* compared to appropriate controls which supports that CLL1CART can effectively reduce LSC burden.

Fig. 8**A****B****C****D****E****F****G**
Extended Data Fig. 10

Healthy control xenografts are required as an additional control to demonstrate specific targeting of JMML and not healthy stem cells.

We have now co-cultured CLL-1 or CD33 CART T cells with healthy control PBMCs for 24 hours at E:T ratio 1:1. We did not observe a reduction in CD34⁺ and CD34⁺CD38⁺ cell numbers with CLL1CART cells but did with CD33CART. This is in line with literature that describe HSPCs expressing CD33 at a higher level compared to CLL-

1¹¹. Due to physiologically very low numbers of CD34⁺CD38⁻ cells in control PB, this population could not be assessed by flow cytometry (Supplementary Data Fig. 2 and lines 181-183 of the manuscript).

We respectfully chose not to perform additional healthy control xenograft studies to demonstrate that healthy stem cells are not targeted as CLL-1 CAR T cells have now been tested *in vitro*^{12,13}, *in vivo* and in human studies, even in pediatric patients, that demonstrate their safety and non-targeting of healthy control HSCs. One study in refractory/relapsed pediatric AML treated four patients in which all experienced manageable and low-grade adverse events¹⁴ (now reference 24 in the manuscript).

Reviewer #3

In the manuscript "Discovery and development of CLL-1 as a cellular immunotherapy target in JMML", the teams of Elliot Stieglitz and Arun P. Wiita identified CLL-1 as a targetable surface marker on JMML cells. They also developed several CLL-1 directed CAR-T cells and showed their efficacy against JMML, both *in vitro* and in a PDX mouse model. The use of immunotherapy in JMML is very interesting, and the combined use of bulk and single cell RNA sequencing, mass spectrometry and flow cytometry represents a powerful approach. Due to the poor outcome of high risk JMML patients, there is a high medical need for novel therapies. However, there are several limitations of this study precluding publication in the current form. The major weakness of the manuscript is the over/misinterpretation of some results, especially the statement that leukemia stem cells (LSC) are successfully hit by CLL1CART. Indeed, none of the experiments performed in this manuscript addressed LSCs or HSCs.

Major points:

- Throughout the manuscript, LSCs were defined as CD34⁺CD38⁺ cells. However, it has been shown earlier that JMML-LSCs are CD34⁺CD38⁻ (Louka et al, JEM, 2021). Results shown in Figure 3D (direct costaining with CD33) and in Figure 4G (indirect evidence by use of CD33CART) indicate that the CD34⁺ cells analyzed by the authors rather are differentiating, myeloid progenitors. The authors should show the expression of CLL-1 on CD34⁺CD38⁻ cells (ideally costained with CD90), demonstrate depletion of these cells by CLL1CART treatment *in vitro* and in PDX.

We agree with the LSC definition as outlined by the reviewer (CD34⁺CD38⁻) from Louka et al⁴ which described the three LSC populations. The reviewer raises an important point about the relevance of CLL-1 expression specifically

on CD34⁺CD38⁻CD90⁺ cells. We assessed two peripheral blood and three bone marrow samples from JMML patients for CD90 expression. We used the same antibody clone as from the Louka et al. paper and were able to successfully titrate it on Jurkat cells – we therefore do not believe that there was a technical problem. Unexpectedly none of the samples had a relevant amount of CD34⁺CD38⁻CD90⁺ cells (see below), we were therefore not able to assess CLL-1 expression on this particular population. It was previously reported that not all JMML patients have a CD34⁺CD38⁻CD90⁺ population¹⁵. Regardless, we believe that the CD34⁺CD38⁻ population encompasses the true LSC population and used this for the ensuing experiments.

Response letter Fig. 2 – not included in manuscript

We followed the reviewer’s suggestion and analyzed our data for the expression of CLL-1 on CD34⁺CD38⁻ cells (Extended Data Fig. 3E-F and page 7 of this document). Our analysis shows heterogenous expression of CLL-1 on CD34⁺CD38⁻ cells with trends towards increased expression in JMML CD34⁺CD38⁻ bone marrow cells (that also have LSC properties as per Louka et al.) versus controls. CLL-1 expression on CD34⁺CD38⁻ cells was more varied compared to controls. We tempered our comments throughout the manuscript (lines 88-90/134/165-166/191-193 of the manuscript) and acknowledge that CLL-1 is not uniformly overexpressed on CD34⁺CD38⁻ LSCs.

As expected, CD34⁺ cells from control peripheral blood did not yield an adequate number of cells for subsetting into CD34⁺CD38⁻ and CD34⁺CD38⁺ cells (Extended Data Fig. 3F).

Demonstration of depletion of CD34⁺CD38⁻ cells by CLL1CART *in vitro* has now been added to Fig. 4G (lines 226-228 of the manuscript):

Demonstration of depletion of CD34⁺CD38⁻ cells by CLL1CART *in vivo* has now been added to Fig. 6F (HM5896 termination study, lines 267-271 of the manuscript) and Fig. 7E/Extended Data Fig. 9F (HM2130 termination study, lines 277-278 of the manuscript):

Fig. 6F:

Fig. 7E:

Extended Data Fig. 9F:

In addition, serial transplantation is required to confirm that JMML-LSCs treated with CLL1CART were successfully depleted. Only such experiments can determine the effects of CLL1CART on LSCs.

Both reviewer 2 and 3 astutely asked for similar experiments to document the effects of CLL1CART on LSCs.

For secondary transplantation, frozen bone marrow cells from the HM2130 survival study were thawed and 1e6 cells/mouse were injected into busulfan-conditioned 11-week-old female NOG-EXL mice (lines 664-667 of the manuscript).

CLL1CART mice had lower JMML burden in peripheral blood over time versus UTD and PBS (Fig. 8B, Extended Data Fig. 10A). This corresponded to significantly reduced spleen length and weight (Fig. 8C) and lower tumor burden in bone marrow, spleen and cardiac blood (Fig. 8D) upon termination 61 days after bone marrow injection as well as lower HSPC burden (Figure 8E-G, Extended Data Fig. 10B-D, see pages 22-23 of this document, lines 295-302 of the manuscript). The drastic reduction in serial passagability of CLL1CART treated cells *in vivo* supports that CLL1CART can successfully deplete JMML LSCs (lines 59-60/103-105/316-320 of the manuscript).

- Introduction: CLL-1 should be introduced including its function, physiological expression and earlier use of CLL1CART.

We thank the reviewer for this suggestion and have added the respective paragraph to the introduction:

Lines 91-101:

“C-type lectin-like receptors play an essential role in regulating innate and adaptive immunity¹⁶. CLL-1 recognizes various ligands on dead cells¹⁷ and specifically recruits inhibitory Src homology region 2 domain-containing phosphatase (SHP)-1 and SHP-2, alleviating inflammation¹⁸. C-type lectin-like molecule-1 (CLL-1), encoded by *CLEC12A*, has emerged as an effective target in the treatment of acute myeloid leukemia (AML), given its high expression on bulk AML cells and LSCs^{12,19}. In normal human hematopoietic tissues, CLL-1 is highly expressed on myeloid progenitor cells²⁰ but is largely absent in normal Lin⁻CD34⁺CD38⁻ HSCs²⁰, offering a unique therapeutic window for targeting myeloid leukemia^{12,21}.

CLL-1 CAR T cells have shown promise against AML *in vitro* and *in vivo*^{12,13}. Clinical trials have indicated tolerability and efficacy of CLL-1 CAR T cell therapy in children^{14,22-24} and adults²⁵ with relapsed or refractory AML. “

- Results: CLL-1 expression on JMML MNCS: the CLL-1 expression is very variable. The authors should include some information on the expression of CLL-1 in the different methylation groups (LM-IM-HM). Since novel therapies are only required for IM/HM JMML patients, this information is crucial.

We thank the reviewer for making this important suggestion. We have now analyzed CLL-1 (*CLEC12A*) expression across JMML methylation groups (low, intermediate, high, unknown) versus controls and visualized the data using

violin plots. Each group was compared to control using an unpaired two-tailed Wilcoxon rank-sum test, revealing significant upregulation in all JMML groups compared to controls ($p < 0.05$ for all comparisons with post hoc Bonferroni correction) (Extended Data Fig. 1D, lines 130-132 of the manuscript). Importantly, the distribution patterns are broadly similar across methylation groups, suggesting uniform upregulation of *CLEC12A* irrespective of methylation status. **We further confirmed this observation using a multinomial logistic regression analysis, which revealed no statistically significant association between *CLEC12A* expression and methylation group membership** ($p = 0.5090$, OR = 1.19 [95% CI: 0.71-2.02] for intermediate; $p = 0.8358$, OR = 0.96 [95% CI: 0.65-1.42] for high methylation groups), both compared to the low methylation group. These odds ratios indicate that the likelihood of a sample belonging to the intermediate or high methylation group does not significantly change with *CLEC12A* expression levels. Together these results demonstrate the uniform expression of *CLEC12A* across methylation groups. This underlines anti-CLL-1 CAR T cell therapy could potentially benefit JMML patients with intermediate and high methylation (i.e., those most likely to require investigational therapies).

- Figure 3A: How was the CLL1-MFI determined? To determine upregulation of an individual cell, the MFI should be determined only in CLL-1 positive cells. If the whole cell population was analyzed, the higher MFI levels simply reflects the higher amount of CLL-1 positive cells. In addition, Figure 3A would benefit from a logarithmic presentation.

We thank the reviewer for the opportunity to clarify how we determined MFI. The CLL-1 MFI as shown in Fig. 3A was determined based on the whole cell population (lines 176-180 of the manuscript). We followed the excellent suggestion to restrict our analysis on just CLL-1+ cells and did not observe differences in the CLL-1 MFI for CLL-1+ cells in bone marrow or peripheral blood (Extended Data Fig. 3C). Therefore, as the reviewer pointed out, the increased CLL-1 MFI of the JMML mononuclear cell samples likely reflects the higher amount of CLL-1+ cells.

We also performed quantification of CLL-1 molecules on the surface of JMML bone marrow and peripheral blood samples with Quantum APC MESF beads (Bangs Laboratories, 823). These beads contain known numbers of fluorophore molecules per bead to generate a standard curve with the median fluorescence intensity (MFI) of the beads for that fluorophore. Using the MFI of the sample the number of CLL-1 molecules per cell was determined. We observed CLL-1 densities above the reported antigen threshold (a few thousand per target cell) for lysis and cytokine production²⁶ (Fig. 3D and lines 184-186 of the manuscript).

We have now changed Fig. 3A to a logarithmic plot.

- Data shown in Figure 4 are only technical replicates. At least 3-4 patient samples should be used to demonstrate that the effect is reproducible. For Figure 4C-D, raw data instead of normalized values should be shown.

We agree with the reviewer that within each individual experiment shown in Fig. 4 there are technical replicates. However multiple patient samples as biological replicates were used to demonstrate efficacy of CLL1CART against primary JMML: Fig. 4C uses HM5867, Fig. 4E uses HM2616 and Fig. 4F-G use HM5862 (lines 213-214 of the manuscript). We therefore believe that we reproducibly showed efficacy of CLL1CART in cytoreducing JMML cells.

For Fig. 4C-D, the raw data instead of normalized values are now shown.

- It is not fully clear, when the PDX mice were treated. Figure 6A and 7A indicate that treatment was initiated 4 weeks after xenotransplantation. This is a very early time point to treat JMML in a PDX, given that JMML results in a slow, chronic disease in xenotransplanted mice. The authors should discuss this.

We agree that our initial wording was not sufficiently clear. The reviewer correctly states that PDX mice were treated 4 weeks after xenotransplantation as shown in Fig. 6A and 7A (now clarified in line 267 of the manuscript).

We employed a threshold of 0.5% or more human CD45⁺ cells in peripheral blood as proof of engraftment upon which randomization and treatment followed. We thereafter followed an accepted convention for PDX²⁷ used in e.g. BCR-ABL lymphoblastic leukemia²⁸ and small molecule intervention trials in JMML²⁹. In our HM2130 survival study this led to all untreated PBS mice reaching humane endpoint after 91 days which is faster and more aggressive than other published JMML PDX models (e.g. >200 days for all untreated mice to reach humane endpoint)³⁰. Mean engraftment [%] for the three PDX trials shown were also higher than 0.5% (Extended Data Fig. 8A, 9A, 9F):

	CLL-1 KO	UTD KO	PBS
HM5896 termination	0.72	0.75	1.22
HM2130 termination	1.24	1.24	1.84
HM2130 survival	1.25	1.30	1.11

Other published PDX studies of other leukemias often inject CAR T cells much earlier (five days after leukemia injection) without any confirmation of engraftment³¹.

Given the rapid engraftment compared to published endpoints of untreated JMML PDX mice and higher mean engraftment than the conventionally used threshold, we believe that we treated our mice at a relatively later timepoint compared to other publications.

- Discussion line 263: please remove the word "specifically". As stated by the authors, CLL1CART were used already for other disease entities (lines 308 etc.).

We thank the reviewer for pointing this out and followed the suggestion.

- Line 287: The authors should omit the statement that CLL1CART is superior over CD33CART. Their *in vitro* data indicate that CD33CART is more potent than CLL1CART (Figure 4), and no direct comparison was performed *in vivo*.

We agree with the reviewer and have removed this wording. We also agree that we have not made any *in vivo* comparison of CLL1CART versus CD33CART and have therefore adjusted the paragraph as follows:

Lines 346-355:

“Although we did not directly compare CLL1CART to other targets *in vivo*, CLL-1 has theoretical advantages over other commonly investigated myeloid targets including CD33, such as lower expression on normal HSPCs³² which may reduce prolonged myeloablation³³. Additionally, in a Phase I/II CD33CART trial in children and young adults with relapsed/refractory AML only 2/19 infused patients achieved complete remission³⁴ indicating room for improvement.

Similar to lymphoid malignancies where multiple antigens including CD19, CD20, and CD22 are being targeted, myeloid leukemias will also likely require targeting of multiple antigens either alone or in combination. It is possible that targeting a combination of targets, such as CLL-1, CD33 and CD123 might be most effective in JMML even though CD33 and CD123 did not pass the filters in our bioinformatic pipeline.”

Minor points:

- Why are JMML cells defined as "GFP-negative"?

We apologize that this was not clear. The gating strategy to identify JMML cells included “GFP-negative” in some experiments where they were co-cultured with GFP-positive CAR T cells to have an additional discriminatory gate

to exclude CAR T cells as much as possible from analysis when focusing on remaining JMML cells (gating strategy now added to Supplementary Data Fig. 14A).

Supplementary Data Fig. 14

- Sentence in line 235-236 is not correct (First, an NRAS-mutant JMML PDX model...).

We thank the reviewer for pointing this out and the manuscript now includes the correct sentence: “First, we tested CLL1CART in an NRAS-mutant JMML PDX model” (lines 264-267 of the manuscript).

1. Willier, S. *et al.* CLEC12A and CD33 coexpression as a preferential target for pediatric AML combinatorial immunotherapy. *Blood* **137**, 1037–1049 (2021).
2. Schippel, N. & Sharma, S. Dynamics of human hematopoietic stem and progenitor cell differentiation to the erythroid lineage. *Exp. Hematol.* **123**, 1–17 (2023).
3. Peters, I. J. A., Pater, E. de & Zhang, W. The role of GATA2 in adult hematopoiesis and cell fate determination. *Front. Cell Dev. Biol.* **11**, 1250827 (2023).
4. Louka, E. *et al.* Heterogeneous disease-propagating stem cells in juvenile myelomonocytic leukemia. *J Exp Medicine* **218**, e20180853 (2021).
5. Andreatta, M., Berenstein, A. J. & Carmona, S. J. scGate: marker-based purification of cell types from heterogeneous single-cell RNA-seq datasets. *Bioinformatics* **38**, 2642–2644 (2022).
6. Liu, Y., Beyer, A. & Aebersold, R. On the Dependency of Cellular Protein Levels on mRNA Abundance. *Cell* **165**, 535–550 (2016).
7. S., N. S. *et al.* Axicabtagene Ciloleucel CAR T-Cell Therapy in Refractory Large B-Cell Lymphoma. *N. Engl. J. Med.* **377**, 2531–2544 (2017).

8. M., K. H. *et al.* Inotuzumab Ozogamicin versus Standard Therapy for Acute Lymphoblastic Leukemia. *N. Engl. J. Med.* **375**, 740–753 (2016).
9. Cohen, A. D. *et al.* B cell maturation antigen-specific CAR T cells are clinically active in multiple myeloma. *J. Clin. Investig.* **129**, 2210–2221 (2019).
10. Boulch, M. *et al.* Tumor-intrinsic sensitivity to the pro-apoptotic effects of IFN- γ is a major determinant of CD4+ CAR T-cell antitumor activity. *Nat. Cancer* **4**, 968–983 (2023).
11. Haubner, S. *et al.* Coexpression profile of leukemic stem cell markers for combinatorial targeted therapy in AML. *Leukemia* **33**, 64–74 (2019).
12. Wang, J. *et al.* CAR-T cells targeting CLL-1 as an approach to treat acute myeloid leukemia. *J Hematol Oncol* **11**, 7 (2018).
13. Tashiro, H. *et al.* Treatment of Acute Myeloid Leukemia with T Cells Expressing Chimeric Antigen Receptors Directed to C-type Lectin-like Molecule 1. *Mol Ther* **25**, 2202–2213 (2017).
14. Zhang, H. *et al.* Anti-CLL1 Chimeric Antigen Receptor T-Cell Therapy in Children with Relapsed/Refractory Acute Myeloid Leukemia Anti-CLL1 CAR T-Cell Therapy in R/R-AML. *Clin Cancer Res* **27**, 3549–3555 (2021).
15. Caye, A. *et al.* Despite mutation acquisition in hematopoietic stem cells, JMML-propagating cells are not always restricted to this compartment. *Leukemia* **34**, 1658–1668 (2020).
16. Chiffolleau, E. C-Type Lectin-Like Receptors As Emerging Orchestrators of Sterile Inflammation Represent Potential Therapeutic Targets. *Front. Immunol.* **9**, 227 (2018).
17. Neumann, K. *et al.* Clec12a Is an Inhibitory Receptor for Uric Acid Crystals that Regulates Inflammation in Response to Cell Death. *Immunity* **40**, 389–399 (2014).
18. Ma, H., Padmanabhan, I. S., Parmar, S. & Gong, Y. Targeting CLL-1 for acute myeloid leukemia therapy. *J. Hematol. Oncol.* **12**, 41 (2019).
19. Bakker, A. B. H. *et al.* C-Type Lectin-Like Molecule-1: A Novel Myeloid Cell Surface Marker Associated with Acute Myeloid Leukemia. *Cancer Res.* **64**, 8443–8450 (2004).
20. Bill, M. *et al.* Mapping the CLEC12A expression on myeloid progenitors in normal bone marrow; implications for understanding CLEC12A-related cancer stem cell biology. *J. Cell. Mol. Med.* **22**, 2311–2318 (2018).
21. Laborda, E. *et al.* Development of A Chimeric Antigen Receptor Targeting C-Type Lectin-Like Molecule-1 for Human Acute Myeloid Leukemia. *Int. J. Mol. Sci.* **18**, 2259 (2017).
22. Zhang, H. *et al.* Characteristics of anti-CLL1 based CAR-T therapy for children with relapsed or refractory acute myeloid leukemia: the multi-center efficacy and safety interim analysis. *Leukemia* **36**, 2596–2604 (2022).
23. Zhang, H. *et al.* Successful Anti-CLL1 CAR T-Cell Therapy in Secondary Acute Myeloid Leukemia. *Frontiers Oncol* **10**, 685 (2020).
24. Pei, K. *et al.* Anti-CLL1-based CAR T-cells with 4-1-BB or CD28/CD27 stimulatory domains in treating childhood refractory/relapsed acute myeloid leukemia. *Cancer Med-us* (2023) doi:10.1002/cam4.5916.

25. Jin, X. *et al.* First-in-human phase I study of CLL-1 CAR-T cells in adults with relapsed/refractory acute myeloid leukemia. *J Hematol Oncol* **15**, 88 (2022).
26. Watanabe, K. *et al.* Target Antigen Density Governs the Efficacy of Anti-CD20-CD28-CD3 ζ Chimeric Antigen Receptor-Modified Effector CD8⁺ T Cells. *J. Immunol.* **194**, 911–920 (2015).
27. Lapidot, T. *et al.* Identification of human juvenile chronic myelogenous leukemia stem cells capable of initiating the disease in primary and secondary SCID mice. *Blood* **88**, 2655–2664 (1996).
28. Notta, F. *et al.* Evolution of human BCR-ABL1 lymphoblastic leukaemia-initiating cells. *Nature* **469**, 362–367 (2011).
29. Wu, Y. *et al.* BH3 mimetics and azacitidine show synergistic effects on juvenile myelomonocytic leukemia. *Leukemia* 1–13 (2023) doi:10.1038/s41375-023-02079-5.
30. Krombholz, C. F. *et al.* Long-term serial xenotransplantation of juvenile myelomonocytic leukemia recapitulates human disease in Rag2^{-/-} γ c^{-/-} mice. *Haematologica* **101**, 597–606 (2016).
31. Mandal, K. *et al.* Structural surfaceomics reveals an AML-specific conformation of integrin β 2 as a CAR T cellular therapy target. *Nat. Cancer* **4**, 1592–1609 (2023).
32. Gill, S. *et al.* Preclinical targeting of human acute myeloid leukemia and myeloablation using chimeric antigen receptor-modified T cells. *Blood* **123**, 2343–2354 (2014).
33. Mardiana, S. & Gill, S. CAR T Cells for Acute Myeloid Leukemia: State of the Art and Future Directions. *Front. Oncol.* **10**, 697 (2020).
34. Shah, N. N. *et al.* CD33 CAR T-Cells (CD33CART) for Children and Young Adults with Relapsed/Refractory AML: Dose-Escalation Results from a Phase I/II Multicenter Trial. *Blood* **142**, 771 (2023).

Reviewer 2:

The new secondary transplant data is interesting and a nice addition. However, it is unclear if any of the mice developed leukemia in secondary transplant. What did the CLL-KO mice die of in survival analysis?

The secondary transplant study was conducted as a termination study. However, two CLL1CART, one UTD and one PBS mouse reached humane endpoint prior to the planned termination on day 61. The CLL1CART mouse had a low leukemia burden (average leukemia burden in BM was 0.5%), compared to the UTD and PBS mice (2.5% and 14.8%, respectively)(Supplementary Fig. 16E). Upon termination, we observed leukemia in UTD KO and PBS mice in peripheral blood and end-study organs (cardiac blood, spleen and bone marrow) as measured by hCD45+ CD14+ and/or CD33+ and/or CLL-1+ cells with flow cytometry whereas CLL1CART had lower amounts of leukemia (Fig. 8B, D).

The flow cytometry data essentially confirm my previous concern that CLL1 is not overexpressed on the surface of JMML HSPC, nor are CLL1 positive cells increased as a proportion of CD34 compartment. I do not think it is correct to state that CLL1 is "frequently overexpressed on LSC" (line 311), the data do not support this. Absolute numbers of CLL1 expressing cells are increased versus control (particularly in PB) due to the expansion of myeloid cells and precursors as part of the disease phenotype. This is why the RNA-seq and MS data identify CLL1 overexpression. Further, CLL1 negative cells from JMML pts engraft mice. Collectively, this does not make CLL1 a particularly compelling target in my view and yet the functional data is convincing.

I read the arguments made by the authors relating to the disconnect between expression of CLL1 and efficacy of CLL1 targeting CARTs. The comparison with CD19 is for me not convincing. For example, it is very clear that CD19 -ve clones are selected for with CD19 directed CART treatments confirming a very clear link to expression of the target.

I do not wish to hold up publication of this work and would suggest that the authors clearly acknowledge that the expression of CLL1 on HSPC is not different to control HSPC, and that CLL1 is only expressed only on a minority of JMML cells in CD34 compartment (not different to control) and only about 50% of MNC (with a lot of inter-patient heterogeneity, so the reason why CART targeting of CLL1 so effectively eliminates all of JMML cell engraftment is uncertain and requires further study; the authors could speculate on the possible explanation in the discussion.

We acknowledge and appreciate the reviewer's comments and have incorporated them in the manuscript (lines 92, 137, 184-186, 196-198, 233, 236, 319 and 346-348). We thank the reviewer for their helpful comments throughout this critical review.

Additional point:

The data in extended Figure 3C is very oddly distributed for MFI data. It looks bimodal with over 1 log difference between the groups, without explanation.

We thank the reviewer for this observation. A possible explanation has been added to the figure legend: “The seemingly bimodal distribution could be due to some samples having MNCs with higher CLL-1 antigen density (also see Fig. 3D).”